# Structural mimicry of UM171 and neomorphic cancer mutants co-opts E3 ligase KBTBD4 for HDAC1/2 recruitment

Zhuoyao Chen [1], Gamma Chi [1], Timea Balo [2,3], Xiangrong Chen [1], Beatriz Ralsi Montes[1], Steven C. Clifford [4], Vincenzo D'Angiolella [5], Timea Szabo [2], Arpad Kiss[2], Tibor Novak[2], András Herner [2], András Kotschy[2] & Alex N. Bullock [1] ✉

Neomorphic mutations and drugs can elicit unanticipated effects that require mechanistic understanding to inform clinical practice. Recurrent indel mutations in the Kelch domain of the KBTBD4 E3 ligase rewire epigenetic programs for stemness in medulloblastoma by recruiting LSD1-CoREST-HDAC1/2 complexes as neo-substrates for ubiquitination and degradation. UM171, an investigational drug for haematopoietic stem cell transplantation, was found to degrade LSD1-CoREST-HDAC1/2 complexes in a wild-type KBTBD4-dependent manner, suggesting a potential common mode of action. Here, we identify that these neomorphic interactions are mediated by the HDAC deacetylase domain. Cryo-EM studies of both wild-type and mutant KBTBD4 capture 2:1 and 2:2 KBTBD4-HDAC2 complexes, as well as a 2:1:1 KBTBD4-HDAC2-CoREST1 complex, at resolutions spanning 2.7 to 3.3 Å. The mutant and drug-induced complexes adopt similar structural assemblies requiring both Kelch domains in the KBTBD4 dimer for each HDAC2 interaction. UM171 is identified as a bona fide molecular glue binding across the ternary interface. Most strikingly, the indel mutation reshapes the same surface of KBTBD4 providing an example of a natural mimic of a molecular glue. Together, the structures provide mechanistic understanding of neomorphic KBTBD4, while structure-activity relationship (SAR) analysis of UM171 reveals analog S234984 as a more potent molecular glue for future studies.

The rapid growth of genomic and chemogenomic data has unveiled unexpected complexity in the mechanisms of action of disease-causing mutations and small molecule drugs. As well as inducing loss or gain of function, mutations and small molecules can impart neomorphic effects that yield a different protein function[1–6]. Mutations in isocitrate dehydrogenases (IDH1/2) drive tumour formation by inducing the neomorphic production of oncometabolite 2-hydroxyglutarate[7].

Conversely, immunomodulatory drugs (IMiDs) drive the neomorphic destruction of zinc finger proteins by establishing molecular glue interactions with cereblon, the substrate receptor of a cullin4-RING E3 ligase complex (CRL4[CRBN])[8–10]. Neo-substrates of this IMiD-bound complex include SALL4, responsible for the teratogenicity of thalidomide, as well as IKZF1 and IKZF3, which account for the anti-tumour effects of IMiDs lenalidomide and pomalidomide[8–10]. Defining the molecular basis

[1]Centre for Medicines Discovery, University of Oxford, Oxford OX3 7FZ, UK. [2]Servier Research Institute of Medicinal Chemistry, Zahony u. 7, H-1031 Budapest, Hungary. [3]Hevesy György Ph.D. School of Chemistry, Eötvös Loránd University, Pázmány Péter sétány 1/A, H-1117 Budapest, Hungary. [4]Wolfson Childhood Cancer Research Centre, Newcastle University Centre for Cancer, Newcastle upon Tyne NE1 7RU, UK. [5]The Institute of Genetics and Cancer, University of Edinburgh, Crewe Road South, Edinburgh EH4 2XU, UK. ✉e-mail: alex.bullock@cmd.ox.ac.uk

of neomorphic function is therefore critical for uncovering mechanisms of pathogenicity, as well as for establishing therapeutic approaches.

The promise of targeting "undruggable" proteins has led to a surge of interest in targeted protein degradation that has yielded the discovery of other CRL4-dependent molecular glues, including degraders of RBM39 that utilise the substrate receptor DCAF15[11] and cyclin K degraders that utilise a drug-induced complex of DDB1 and CDK12[12,13]. The targeted protein degradation paradigm was recently expanded by the discovery of neomorphic mutations in an E3 ligase[14]. Kelch repeat and BTB domain-containing protein 4 (KBTBD4) is the substrate receptor of a cullin3-dependent E3 ligase complex (CRL3[KBTBD4]). Recurrent in-frame insertions in KBTBD4 have been reported as putative driver mutations in group 3 and group 4 medulloblastomas[15], as well as in pineal parenchymal tumours with intermediate differentiation (PPTID)[16]. These cancer hotspot mutants were found to elicit neomorphic ubiquitination that promoted the degradation of the CoREST transcriptional repressor complex, as well as the increased stemness of medulloblastoma cancer cells[14]. All mutations clustered across just six amino acids within the KBTBD4 Kelch domain. The indel mutation p.R313delinsPRR (herein R313PRR) was particularly prevalent across both brain tumour types and demonstrated a strong neomorphic effect[14-17].

Haematopoietic stem cell (HSC) transplants save the lives of cancer patients, but suffer from the limited number of progenitor cells in umbilical cord blood[18]. UM171 was discovered as a small molecule agonist of HSC expansion[19] and shows promise as an ex-vivo drug treatment for HSC transplants in phase 2 clinical trials[20,21]. UM171 was found to induce degradation of the CoREST complex in a wild-type KBTBD4 and proteasome-dependent manner[22-24], mirroring the neomorphic activity of mutant KBTBD4[14]. Our understanding of these important neomorphic effects is limited by a lack of knowledge of the direct interactors of KBTBD4 and their molecular mechanisms. CoREST proteins function as a scaffold for assembly of a transcriptional repressor complex that contains CoREST1, 2 or 3, the lysine specific demethylase LSD1, histone deacetylase HDAC1 or 2 and various DNA-associated factors[25,26]. Here, we identify HDAC1/2 as the direct interaction partner of KBTBD4 through cellular, biophysical and structural investigations. Cryo-EM structures of wild-type and mutant KBTBD4-HDAC2 complexes confirm UM171 as a bona fide molecular glue and reveal structural mimicry between UM171 and mutant KBTBD4. These data reveal a shared mechanism of neo-substrate recruitment by an E3 cancer mutant and small molecule drug, as well as structural models to guide future drug development for cancer treatment.

## Results

### CoREST complex recruitment requires the full-length KBTBD4[R313PRR] dimer

BTB-Kelch family proteins are known to recruit their substrates through the Kelch domain. In agreement, the KBTBD4 Kelch domain harbours the cancer hotspot mutations that confer neo-substrate interaction. To test whether the Kelch domain alone was sufficient for KBTBD4 mutant binding to CoREST complex, we expressed FLAG-tagged KBTBD4[R313PRR] Kelch domain and full-length constructs and performed FLAG immunoprecipitations in HEK293 cells. Surprisingly, the Kelch domain failed to replicate the neo-substrate binding observed with full-length KBTBD4[R313PRR], indicating that additional domains of KBTBD4 were necessary for binding (Fig. 1a).

BTB-Kelch proteins dimerize through the BTB domain to present two Kelch domains for substrate recruitment. Full-length KBTBD4[R313PRR] may therefore be necessary to establish bivalent substrate interactions involving both Kelch domains within the KBTBD4 dimer. To test this hypothesis, we engineered a monomeric form of the protein by introducing mutations predicted by previous work to disrupt the BTB dimer interface[27], and confirmed its monomeric state using size-exclusion chromatography (Fig. 1b). FLAG immunoprecipitations were then performed comparing the monomeric FLAG-KBTBD4[R313PRR] construct to dimeric controls. Consistent with previous work[14], the full-length KBTBD4[R313PRR] dimer was able to immunoprecipitate CoREST1, HDAC2 and LSD1 from the endogenous CoREST complex (Fig. 1c). By contrast, the monomeric KBTBD4[R313PRR] construct lost binding to all three proteins similarly to the KBTBD4[WT] dimer (Fig. 1c). Taken together, these results suggest that CoREST complex recruitment to KBTBD4 requires both a dimeric BTB domain and a Kelch domain harbouring a neomorphic mutation.

### KBTBD4[R313PRR] is recruited to CoREST N-terminal domains

Our previous proteomics study identified all three CoREST isoforms binding to KBTBD4[R313PRR], but not alternative LSD1 or HDAC1/2-associated scaffolding proteins, such as SIN3A or MTA1[28,29], suggesting that CoREST proteins may be important for KBTBD4 binding. Cryo-EM studies of the full-length LSD1-CoREST1-HDAC1 complex showed that the dynamic features of the complex limited the structure resolution and affected assay readouts[25]. We therefore reasoned that shorter CoREST1 constructs and subcomplexes may be desirable for the further characterization of the structural mechanisms of neomorphic KBTBD4.

The N and C-terminal regions of CoREST1 are known to bind to HDAC1/2 and LSD1, respectively. To map the minimal region required

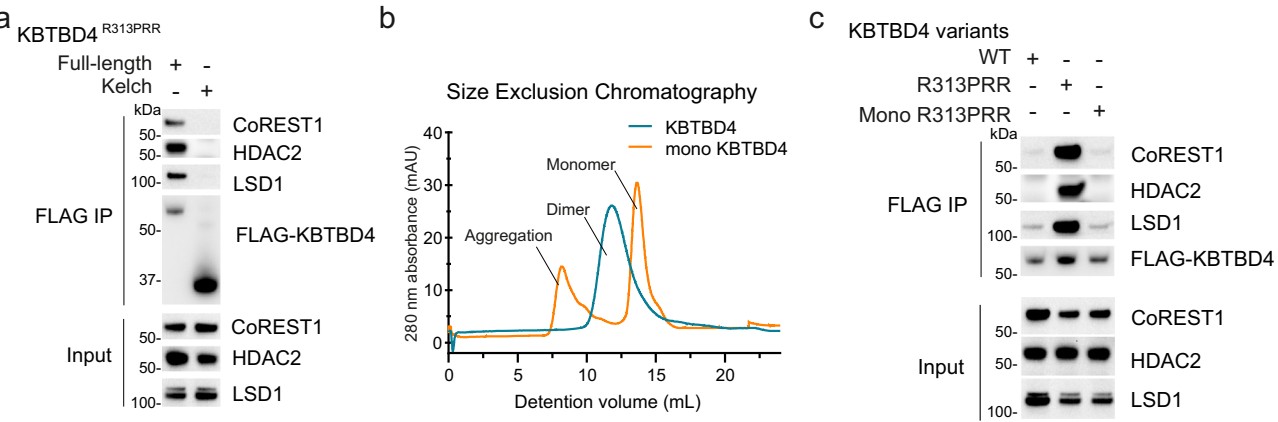

**Fig. 1 | CoREST complex recruitment requires the full-length KBTBD4[R313PRR] dimer. a** Representative immunoblots of lysates and anti-FLAG immunoprecipitates from HEK293 cells expressing FLAG-KBTBD4[R313PRR] full-length or Kelch domain protein. Immunoprecipitation was performed twice with similar results. Source data in this figure are provided in the Source Data file. **b** Size-exclusion chromatography showing oligomerization states of KBTBD4[R313PRR] and its monomeric variant (mono KBTBD4). **c** Representative immunoblots of lysates and anti-FLAG immunoprecipitates from HEK293 cells expressing FLAG-KBTBD4 variants as indicated. Immunoprecipitation was performed twice with similar results. Source data in this figure are provided in the Source Data file.

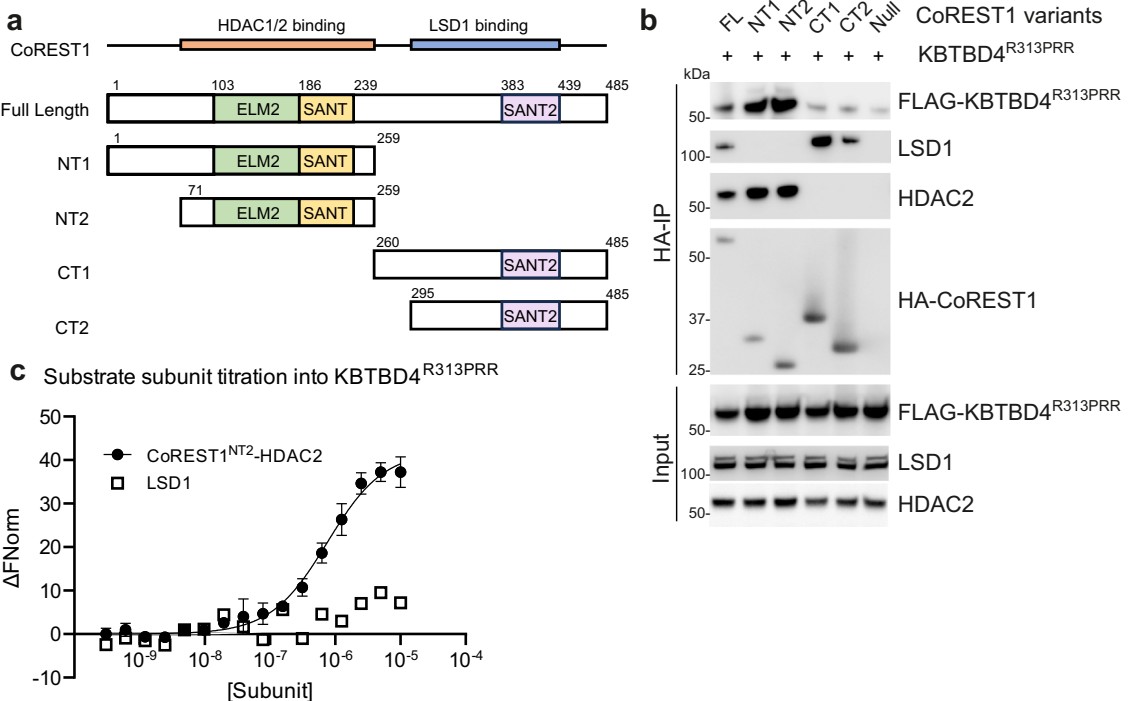

**Fig. 2 | KBTBD4$^{R313PRR}$ is recruited to CoREST N-terminal domains. a** Domain organization of CoREST1 indicating the construct designs used for interaction site mapping. **b** Representative immunoblots of lysates and anti-HA immunoprecipitates from HEK293 cells expressing HA-CoREST1 fragments, as well as FLAG-KBTBD4$^{R313PRR}$ as indicated. Immunoprecipitation was performed at least twice with similar results and source data are provided in the Source Data file. **c** For microscale thermophoresis (MST) experiments, unlabelled CoREST1$^{NT2}$-HDAC2 or LSD1 was titrated at indicated concentrations into 40 nM fluorescently-labelled KBTBD4$^{R313PRR}$. CoREST1$^{NT2}$-HDAC2 bound to KBTBD4$^{R313PRR}$ with a dissociation constant ($K_D$) of 0.75 μM. All MST experiments were performed in triplicates. Data points and error bars represent the means and standard deviations. Source data are provided in the Source Data file.

for KBTBD4$^{R313PRR}$ interaction, we cloned four fragments of CoREST1 designed to explore each terminus with different truncations (Fig. 2a). These HA-tagged constructs were co-transfected into HEK293 cells with KBTBD4$^{R313PRR}$ for HA co-immunoprecipitation. Strikingly, we observed KBTBD4$^{R313PRR}$ binding to the N-terminal CoREST1 fragments NT1 and NT2, but not to the C-terminal fragments CT1 and CT2 (Fig. 2b). The conformational integrity of the N and C-terminal fragments was confirmed by their binding to endogenous HDAC2 and LSD1, respectively (Fig. 2b). These domain mapping experiments demonstrate that the CoREST1 N-terminal region that assembles with HDAC1/2 is required for its recruitment to KBTBD4$^{R313PRR}$.

We selected the smaller CoREST1$^{NT2}$ fragment (a.a. 71-259) for in vitro binding studies as it faithfully spanned the ELM2 and SANT1 domains needed for HDAC1/2 interaction. A previous structure of the homologous MTA1-HDAC1 complex revealed that HDAC1 was enveloped by a flexible arm in the ELM2 domain that would likely be disordered on its own[29]. We, therefore, co-expressed the CoREST1$^{NT2}$ fragment with full-length HDAC1 or 2 for recombinant protein production in Expi293F™ cells. The CoREST1$^{NT2}$-HDAC2 complex showed a higher yield and was purified for affinity measurements using microscale thermophoresis (MST). The resulting CoREST1$^{NT2}$-HDAC2 complex bound strongly to full-length KBTBD4$^{R313PRR}$ with a dissociation constant ($K_D$) value of 0.75 μM (Fig. 2c), consistent with the cellular domain mapping. In contrast, LSD1 alone showed no binding (Fig. 2c), suggesting its co-precipitation with KBTBD4$^{R313PRR}$ in cells was mediated by the CoREST1-HDAC2 subcomplex and not through its direct interaction.

## CoREST1$^{NT2}$-HDAC2 is a common binding entity for KBTBD4$^{R313PRR}$ and KBTBD4$^{WT}$-UM171

We hypothesised that the small molecule drug UM171 would recruit wild-type KBTBD4 to CoREST complexes with a similar mode of action to the R313PRR mutation. We, therefore, purified the full-length wild-type and mutant KBTBD4 proteins and tested their binding to the CoREST1$^{NT2}$-HDAC2 complex in the presence or absence of UM171. We first explored crosslinking of E3-neo-substrate complexes using bis-sulfosuccinimidyl suberate (BS3), which has lysine-reactive hydroxysuccinimide ester groups separated by a spacer arm length of 11.4 Å. Crosslinked and control samples were analyzed on SDS-PAGE to observe mass shifts indicative of complex formation (Fig. 3a). The crosslinked KBTBD4$^{R313PRR}$ control sample exhibited a clear shift to a ~250 kD E3-neo-substrate complex, validating the assay conditions. Encouragingly, an identical mass shift was observed for the mixture of UM171, KBTBD4$^{WT}$ and CoREST1$^{NT2}$-HDAC2 (Fig. 3a), but not when UM171 was absent, showing that the same E3-neo-substrate complex was induced by the small molecule.

Binding affinity measurements of the UM171-induced complex were then performed by MST assay. Titration of the CoREST1$^{NT2}$-HDAC2 complex into KBTBD4$^{WT}$ in presence of 25 μM UM171 produced a dose-dependent interaction estimated to be one order of magnitude weaker ($K_D$ ~ 15 μM) than for the equivalent KBTBD4$^{R313PRR}$ complex, whereas KBTBD4$^{WT}$ without UM171 exhibited smaller fluorescence deviations equivalent to background noise (Fig. 3b). Reassuringly, the same binding was observed when performing the reverse titration of UM171 into a mixture of KBTBD4$^{WT}$ and CoREST1$^{NT2}$-HDAC2 (Fig. 3c). Altogether, these experiments demonstrate that the CoREST1$^{NT2}$-HDAC2 complex is a common binding entity for KBTBD4$^{R313PRR}$, as well as for KBTBD4$^{WT}$ in the presence of UM171.

Neo-substrates recruited to CRL3$^{KBTBD4}$ complexes are known to be ubiquitinated[14]. We therefore reconstituted an in vitro ubiquitination assay using our purified CoREST1$^{NT2}$-HDAC2 substrate, as well as other purified E1, E2 and E3 components. Initial experiments using KBTBD4$^{R313PRR}$ revealed poly-ubiquitin chain formation in the presence of the processive UBE2D family E2s (Fig. 3d). Further investigation

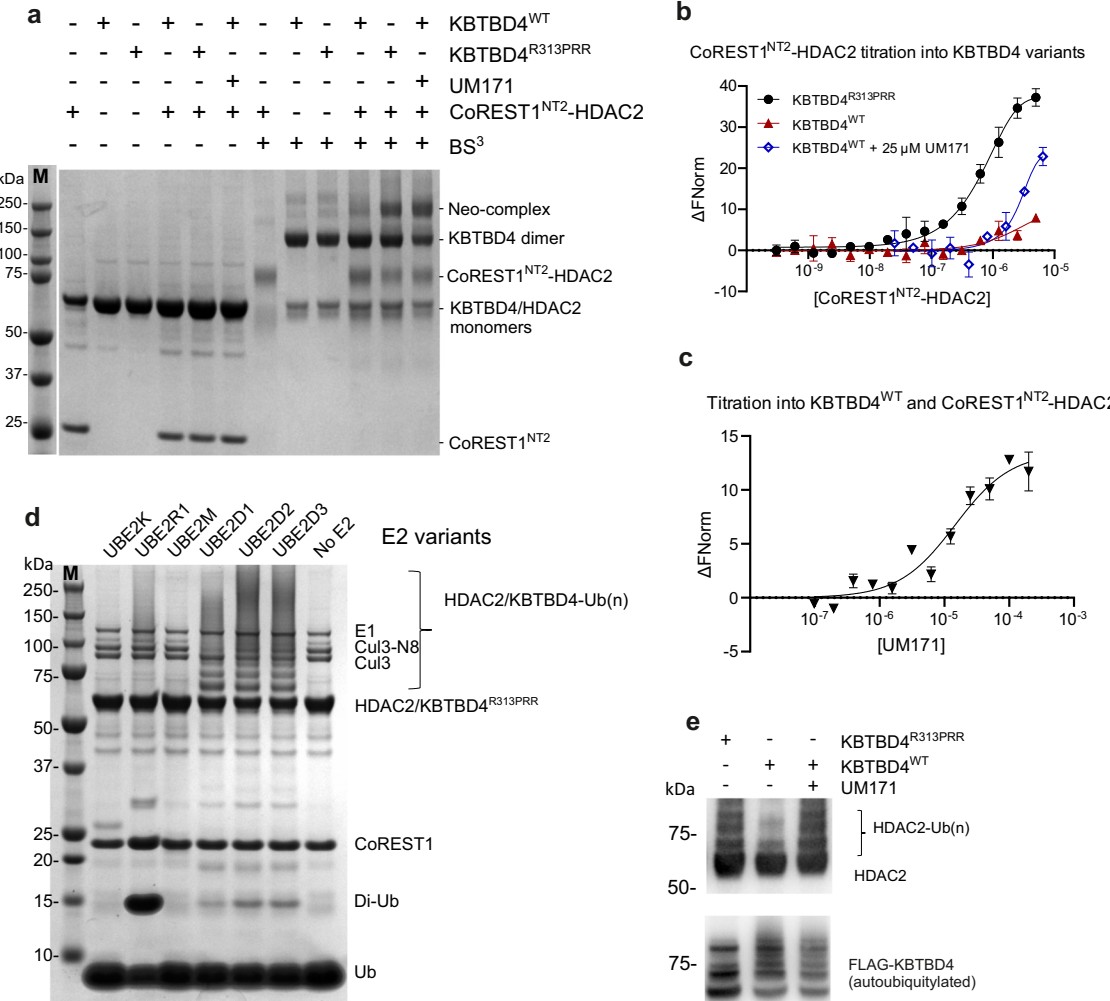

**Fig. 3 | CoREST1^NT2-HDAC2 is the common binding entity for KBTBD4^R313PRR and KBTBD4^WT-UM171.** **a** For crosslinking experiments, 2.5 µM of indicated proteins were mixed and incubated with 4 mM BS³ crosslinker. The reactions were terminated by laemmli buffer and then analyzed in parallel with uncrosslinked controls on an SDS-PAGE gel, followed by Coomassie staining. Crosslinked or uncrosslinked species were labelled alongside the gel image. **b** Unlabelled CoREST1^NT2-HDAC2 was titrated at indicated concentrations into 40 nM fluorescently-labelled KBTBD4 variants, with or without 25 µM UM171, for MST measurements. The apparent $K_D$ values of CoREST1^NT2-HDAC2 binding to KBTBD4^R313PRR and KBTBD4^WT-UM171, respectively, were determined to be 0.75 µM and 15 µM, while negligible binding was observed to KBTBD4^WT alone. **c** Unlabelled UM171 was titrated at indicated concentrations into 40 nM fluorescently-labelled KBTBD4^WT with 5 µM CoREST1^NT2-HDAC2 for MST measurements (apparent $K_D = 14$ µM). All MST experiments were performed in triplicates. Data points and error bars represent the means and standard deviations. Source data are provided in the Source Data file. **d** Various E2 enzymes were mixed with CoREST1^NT2-HDAC2, KBTBD4^R313PRR, neddylated CUL3, E1 and ubiquitin, and incubated in the presence of ATP and MgCl₂ to measure ubiquitylation. The reactions were analyzed on an SDS-PAGE gel and visualised by Coomassie stain. Ubiquitylated HDAC2 and KBTBD4^R313PRR showed higher molecular weight ladders as labelled in the figure. **e** Immunoblots of in vitro ubiquitylation reaction mixtures using UBE2D1 E2 enzyme. KBTBD4 variants and UM171 were added to the reaction mixtures as indicated. The ubiquitylation experiments were performed at least twice with similar results. Source data are provided in the Source Data file.

confirmed neomorphic ubiquitination of HDAC2 by KBTBD4^R313PRR, as well as by KBTBD4^WT dependent on the presence of UM171 (Fig. 3e). By contrast, auto-ubiquitination of KBTBD4 was observed in all samples independently of UM171 (Fig. 3e). Based on these collective data, we selected the CoREST1^NT2-HDAC2 complex as a suitable neo-substrate for Cryo-EM studies.

## Cryo-EM structures reveal HDAC2 as the direct interactor of UM171 and neomorphic mutants of KBTBD4

Structural analysis of KBTBD4-neo-substrate complexes was pursued to define how cancer mutations and UM171 could similarly co-opt KBTBD4 for neomorphic binding to CoREST1^NT2-HDAC2. Cryo-EM data obtained for the complexes of wild-type and mutant KBTBD4 yielded maps with nominal resolutions of 2.7 to 3.3 Å and revealed KBTBD4-neo-substrate complexes with both 2:1 and 2:2 stoichiometry (Fig. 4, Table 1 and Supplementary Figs. 1–3). The best map at

2.7 Å was obtained for the higher affinity sample mixing KBTBD4^R313PRR and CoREST1^NT2-HDAC2. The modelled structure revealed a KBTBD4^R313PRR dimer engaging two HDAC2 molecules in C2 symmetry (2:2 stoichiometry, Fig. 4a, b, although a rare 2D class with one HDAC2 (2:1) was also observed, Supplementary Fig. 4). The mutant KBTBD4^R313PRR was resolved to near full length, consisting of BTB, BACK and Kelch domains, whereas only the histone deacetylase domain could be modelled from the full-length HDAC2 construct (Fig. 4b, c); density was also lacking for its associated CoREST1^NT2 subunit. To determine the structure of the lower affinity KBTBD4^WT complex mediated by UM171, we used the BS3 crosslinker to stabilise the binding of the CoREST1^NT2-HDAC2 neo-substrate. This sample yielded 2:2 and 2:1 KBTBD4^WT-UM171-HDAC2 complexes at 2.9 Å and 3.1 Å, respectively. A subset of the 2:1 complexes showed additional density of lower quality for a CoREST1^NT2 fragment wrapping around HDAC2 at 3.3 Å (Fig. 4d–f). No such density was found in the 2:2

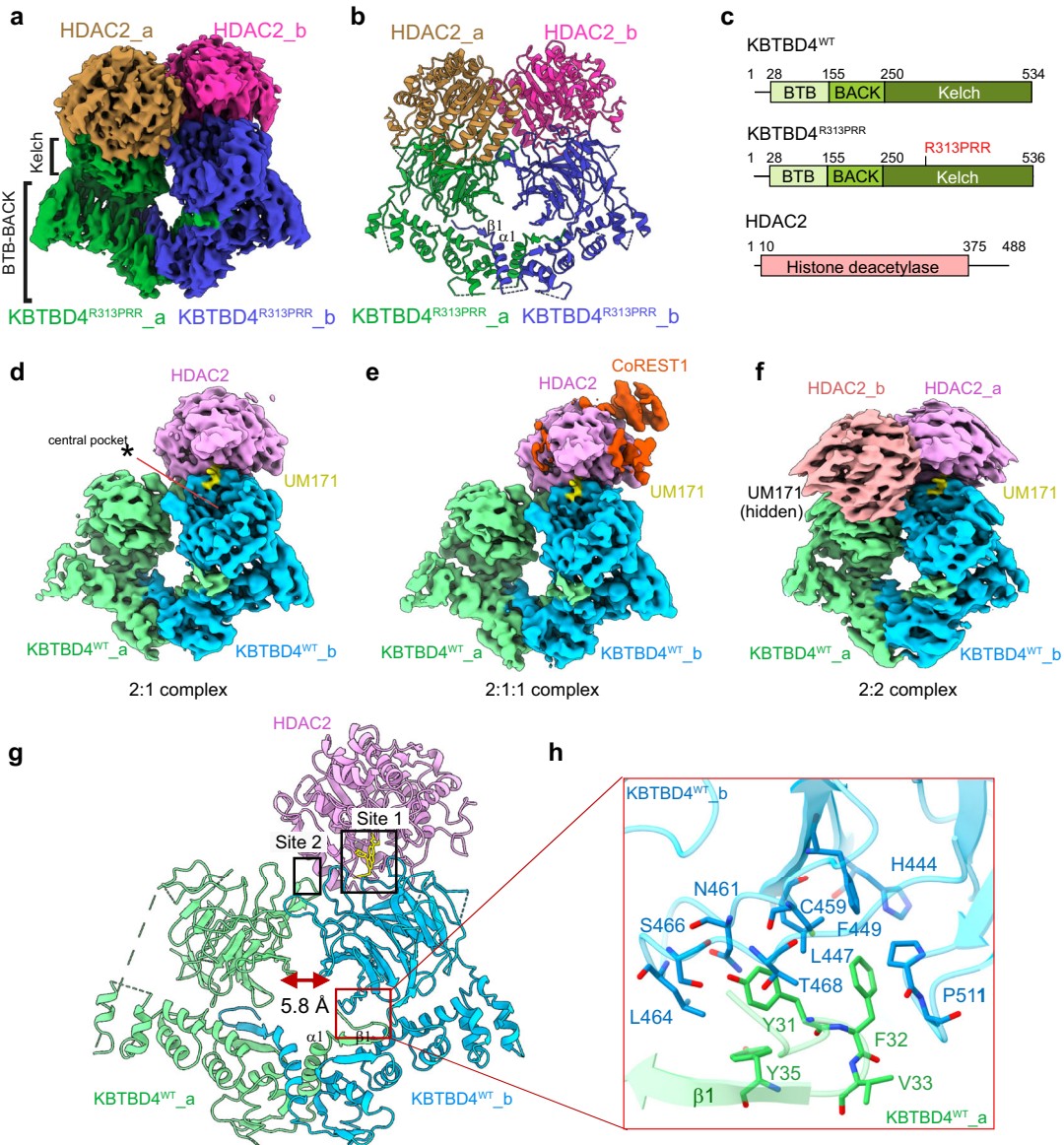

**Fig. 4 | Cryo-EM structures reveal HDAC2 as the direct interactor of UM171 and neomorphic mutants of KBTBD4.** **a** Cryo-EM map of the 2:2 KBTBD4$^{R313PRR}$-HDAC2 complex coloured by individual subunit (contour level = 0.15). BTB-BACK and Kelch regions in KBTBD4 are labelled. **b** Ribbon representation of the 2:2 KBTBD4$^{R313PRR}$-HDAC2 structure with each subunit coloured as in **a**. The dimer interface α1 and β1 in the KBTBD4 BTB domain are labelled. **c** Domain organizations of KBTBD4$^{WT}$, KBTBD4$^{R313PRR}$ and HDAC2. **d**–**f** Cryo-EM maps of UM171-induced KBTBD4$^{WT}$-substrate complexes at 2:1 (KBTBD4:HDAC2, contour level = 0.08), 2:1:1 (KBTBD4:HDAC2:CoREST1, contour level = 0.08) and 2:2 (KBTBD4:HDAC2, contour level = 0.08). UM171 is coloured in yellow. Asterisk indicates the unoccupied central

pocket of the KBTBD4 Kelch domain. **g** Ribbon representation of the KBTBD4$^{WT}$ homodimer binding to UM171 and HDAC2. Subunits are coloured as in **d**–**f**. Each HDAC2 subunit interacts with residues at sites 1 and 2 in the KBTBD4 dimer as boxed and labelled by text. The distance between the two Kelch domains is measured by the side chains of E381. The interaction interface between the N-terminus of KBTBD4 chain a and the Kelch domain of KBTBD4 chain b is indicated with a red box. **h** Close-up view of the interaction interface between the N-terminus of KBTBD4 chain a and the Kelch domain of KBTBD4 chain b. KBTBD4 chains are coloured as in **g** and shown in ribbon representation. Key interacting residues are shown in sticks.

complex, likely due to steric constraints that would prevent CoREST1 binding in this higher order assembly (Supplementary Fig. 5). Importantly, all 3D models identified HDAC2 as the direct interactor of KBTBD4, consistent with crosslinking and MST data showing that HDAC2 was sufficient for KBTBD4 recruitment (Supplementary Fig. 6).

In all structures, each HDAC2 subunit was bound by both Kelch domains in the KBTBD4 dimer (Fig. 4g), further highlighting the importance of dimeric assembly in the BTB protein class[30,31]. Of note, the high-quality tracing of the full-length KBTBD4 chains provided additional insights into the dimerization mechanism. The N-terminal BTB domain dimerized as expected through the α1 helix and a domain-

swapped β1 strand (Fig. 4b, g). However, the BACK domain of KBTBD4 was notably two helices shorter than previously observed for other family members (Supplementary Fig. 7)[30,32–34], allowing closer proximity of the two Kelch domains. Indeed, the Kelch domains in the KBTBD4 dimer were separated by just 5.8 Å (Fig. 4g), in contrast to the 26 Å and 54 Å observed in the cryo-EM structures of KLHL22 and KBTBD2, respectively[32–34]. The more central position of the Kelch domains in KBTBD4 was further stabilised by hydrophobic interaction with an N-terminal extension of the domain-swapped β1 strand (Fig. 4g, h), showing an example of conformational crosstalk between the BTB and Kelch domains of different protomers. These structural features illustrate how each domain in KBTBD4 contributes to neo-

**Table 1 | Cryo-EM data collection and refinement statistics**

| | KBTBD4^WT·UM171-neo-substrate complex | | | KBTBD4^R313PRR-neo-substrate complex |
|---|---|---|---|---|
| Microscope | Titan Krios | | | Titan Krios |
| Detector | K3 | | | K3 |
| Voltage (kV) | 300 | | | 300 |
| Magnification | 105,000 | | | 105,000 |
| Collection mode | Counting | | | Counting |
| Electron exposure (e/Å$^2$) | 38.0 | | | 38.6 |
| Number of frames | 40 | | | 40 |
| Pixel size (Å) | 0.832 | | | 0.832 |
| Defocus range (μm; steps) | −1.0 to -3.2 (0.2) | | | −1.0 to −2.8 (0.2) |
| Number of movies | 12,722 | | | 13,131 |
| Initial Number of particles | 13,588,054 | | | 19,876,414 |
| Number of particles after 2D classification | 511,570 | | | 470,242 |
| Stoichiometry (KBTBD4:HDAC2:CoREST1) | 2:1:0 | 2:1:1 | 2:2:0 | 2:2:0 |
| Number of particles used for 3D refinement | 264,324 | 105,484 | 356,933 | 295,217 |
| Symmetry | C1 | C1 | C2 | C2 |
| Map resolution (Å; FSC threshold = 0.143) | 3.1 | 3.3 | 2.9* | 2.7** |
| Resolution range (Å) | 2.7 – 47.7 | 2.9 – 53.2 | 2.5 – 43.8 | 1.8 – 42.7 |
| Map sharpening B-factor (Å$^2$) | −123.7 | −85.1 | −122.7 | −107.7 |
| Model resolution(Å; FSC threshold = 0.5) | 3.5 | 3.7 | 3.7 | 3.2 |
| Q-score | 0.479 | 0.422 | 0.424 | 0.484 |
| Non-hydrogen atoms | 9006 | 9912 | 9936 | 11450 |
| Protein residues | 1233 | 1381 | 1434 | 1566 |
| Ligands | 2 | 2 | 4 | 2 |
| R.M.S.D | | | | |
| Bond lengths (Å) | 0.004 | 0.001 | 0.003 | 0.005 |
| Bond angles (°) | 0.749 | 0.398 | 0.633 | 0.803 |
| Validation | | | | |
| Molprobity score | 1.93 | 1.78 | 1.86 | 2.10 |
| Clash score | 13.85 | 5.73 | 5.19 | 11.91 |
| Rotamer outliers (%) | 0.24 | 2.25 | 1.52 | 3.07 |
| Ramachandran plot | | | | |
| Favoured (%) | 95.91 | 96.76 | 92.98 | 97.22 |
| Allowed (%) | 4.09 | 3.24 | 7302 | 2.78 |
| Disallowed (%) | 0 | 0 | 0 | 0 |
| EMDB Code | 51335 | 51336 | 51338 | 51337 |
| PDB Code | 9GGL | 9I2C | 9GGN | 9GGM |

*FSC-calculated nominal resolution. Practical resolution on manual inspection is closer to ~4 Å.

**FSC-calculated nominal resolution. Practical resolution on manual inspection is closer to ~3.5 Å.

substrate recruitment and explains the requirement for dimerization in the cellular binding data shown in Fig. 1.

### Structural basis for neomorphic substrate recruitment by the cancer mutant KBTBD4^R313PRR

The structure of the mutant complex shows that the Kelch domain of KBTBD4^R313PRR retains a conserved β-propeller fold of six blades (I to IV) despite harbouring the R313PRR indel mutation. Each blade consists of four antiparallel beta strands (βA to βD), connected by DA and BC loops of variable length. The recurrent cancer mutation R313PRR lies within the BC loop of blade II (BC II) (Fig. 5a), in agreement with previous predictions[14,15].

Each HDAC2 subunit in the 2:2 model makes direct contacts with both Kelch domains in the KBTBD4^R313PRR dimer by engaging the mutant BC II loop from one chain (site 1) and the BC IV loop (blade IV, site 2) from the other (Fig. 5a, b). Unexpectedly, both sites involve interactions at the outer rim of the Kelch propeller (Fig. 5b), rather than binding to the central pocket, as reported for many other BTB-Kelch proteins[35–40]. In contrast to the dimeric

assembly of KBTBD4, no apparent intermolecular interaction is observed between the two HDAC2 chains, suggesting that the two HDAC2 recruitment events are likely independent. Given the applied C2 symmetry in the model, only one chain of HDAC2 is discussed in further structural analysis.

Site 1 involving the mutant BC II loop forms the major interaction site for HDAC2 recruitment. Here, the arginine cluster introduced by the KBTBD4^R313PRR mutation and preceding R312 establish both polar and hydrophobic interactions (Fig. 5c–f). Most importantly, the indel mutation extends the length of BC II and shifts the position of KBTBD4 R312, a non-mutated residue, by 5.9 Å to the tip of the loop. This structural alteration allows the R312 side chain to insert into the catalytic pocket of HDAC2 to within 5.7 Å of the catalytic Zn$^{2+}$ ion, which natively binds to acetyl-lysines in histone H3 and H4 tails[41–43]. Similar to acetyl-lysine binding[44,45], KBTBD4^R313PRR R312 engages HDAC2 Y304 through a hydrogen bond interaction that induces an 'in' position of Y304, as typically observed in active class I HDACs (Fig. 5d and Supplementary Fig. 8a, b). In addition, the R312 side chain is sandwiched between HDAC2 F151 and F206 to form

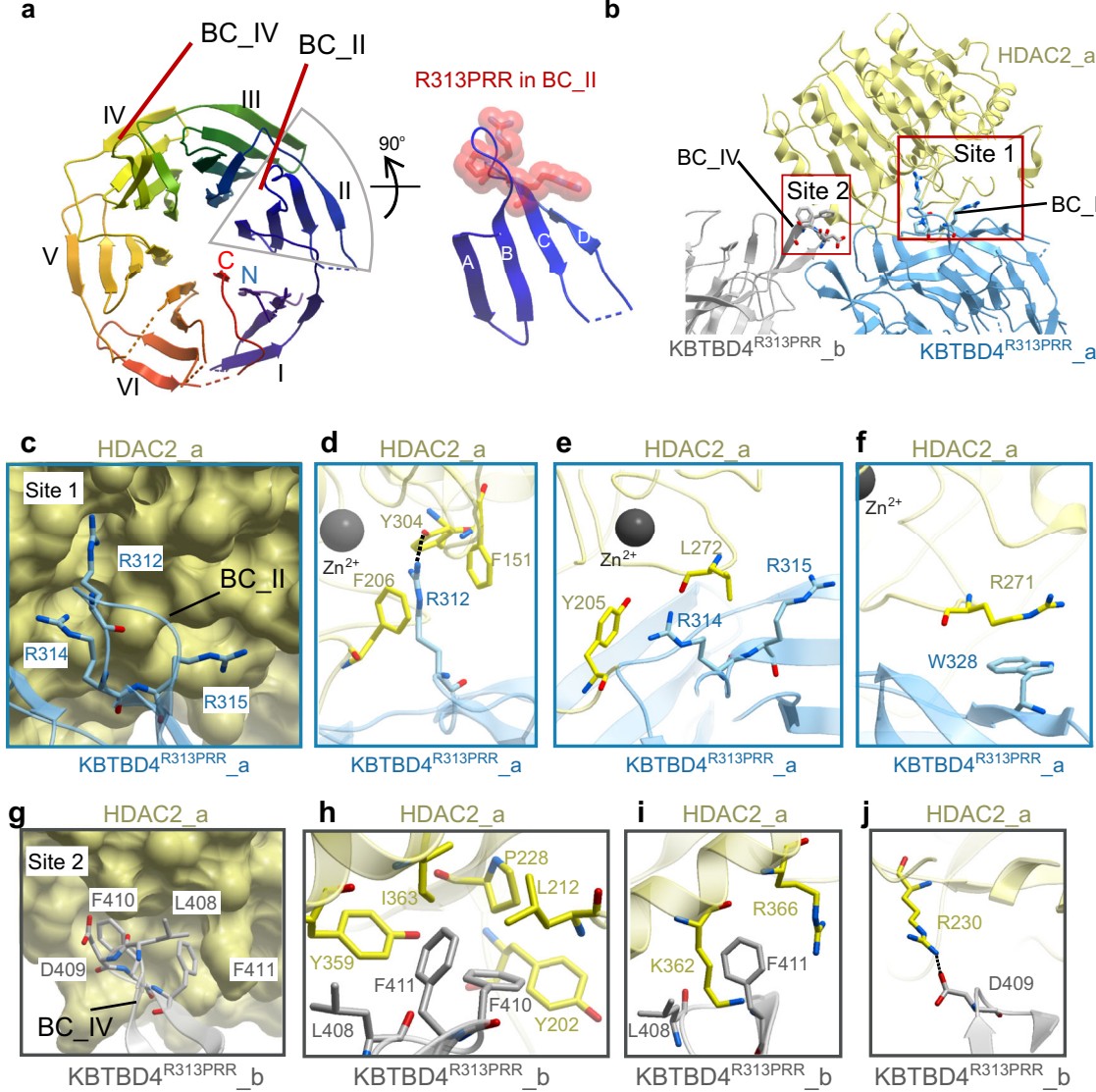

**Fig. 5 | Cryo-EM structure of the KBTBD4^R313PRR-HDAC2 complex reveals the basis of neomorphic substrate recruitment in cancer. a** Ribbon representation of the KBTBD4^R313PRR Kelch domain. The Kelch β-propeller consists of blades I to VI with each blade consisting of four antiparallel β strands (labelled A-D). Loops connecting βB and βB in blade II and IV are labelled as BC II and BC IV, respectively. The R313PRR cancer mutation in BC II is shown as red spheres. **b** Ribbon representation of HDAC2 binding to KBTBD4^R313PRR_a at site 1 (BC II) and KBTBD4^R313PRR_b at site 2 (BC IV). The second HDAC2 chain in the model is a symmetric duplication of the first and, therefore, not shown for simplicity. **c** Close-up view of site 1 interface.

HDAC2 is shown as yellow surface and KBTBD4^R313PRR_a as cyan ribbon. The key interacting arginine residues in KBTBD4^R313PRR_a are shown as sticks in cyan. **d**–**f** Close-up views of site 1 key interfacial residues shown in sticks with the same colour scheme as **c**. Hydrogen bonding is indicated by dashed lines. **g** Close-up view of the site 2 interface. HDAC2 is shown as yellow surface and KBTBD4^R313PRR_b as grey ribbon. The key interacting residues in KBTBD4^R313PRR are shown as grey sticks. **h**–**j** Close-up views of site 2. Key interfacial residues are shown in sticks with the same colour scheme as in **g**.

cation-π interactions (Fig. 5d). Additional cation-π interaction is observed between KBTBD4^R313PRR R314 and HDAC2 Y205 close to the exit of the HDAC2 catalytic pocket (Fig. 5e). The aliphatic portions of KBTBD4^R313PRR R314 and R315 also form hydrophobic interactions with HDAC2 L272 (Fig. 5e), while the preceding HDAC2 R271 forms a parallel π-stacking interaction with KBTBD4^R313PRR (Fig. 5f). The benzamide HDAC1/2 inhibitor BRD6929 is known to insert into the same HDAC2 catalytic site[46,47] and would be predicted to compete with this binding mode (Supplementary Fig. 8c, d). Indeed, the addition of BRD6929 abrogated all KBTBD4^R313PRR-HDAC2 binding in the MST assay (Supplementary Fig. 8e), further highlighting the importance of these pocket interactions. Overall, the site 1 contacts demonstrate the neomorphic interactions facilitated by the KBTBD4^R313PRR cancer mutation, as well as the native surface compatibility of surrounding wild-type residues.

The site 2 interface is comprised entirely of non-mutated residues. Here, KBTBD4^R313PRR L408, F410 and F411 from the BC IV loop form extended hydrophobic interactions with an HDAC2 cleft that features Y202, L212, P228, Y359 and I363 (Fig. 5g, h). Outside of this cleft, KBTBD4^R313PRR L408 and F411 also interact with the aliphatic parts of HDAC2 K362 and R366 (Fig. 5i). Additionally, the binding is stabilised by a salt bridge between KBTBD4^R313PRR D409 and HDAC2 R230 (Fig. 5j). These interactions suggest a substantial contribution of the native site 2 in KBTBD4^R313PRR to HDAC2 recruitment, demonstrating additional surface complementarity outside the cancer hotspot mutation site.

**UM171 acts as a molecular glue to recruit HDAC2 to KBTBD4^WT**
The cryo-EM datasets for the KBTBD4^WT-UM171-HDAC2 complexes yielded 2:1 and 2:2 models displaying an RSMD value of 1.55 Å across 1055

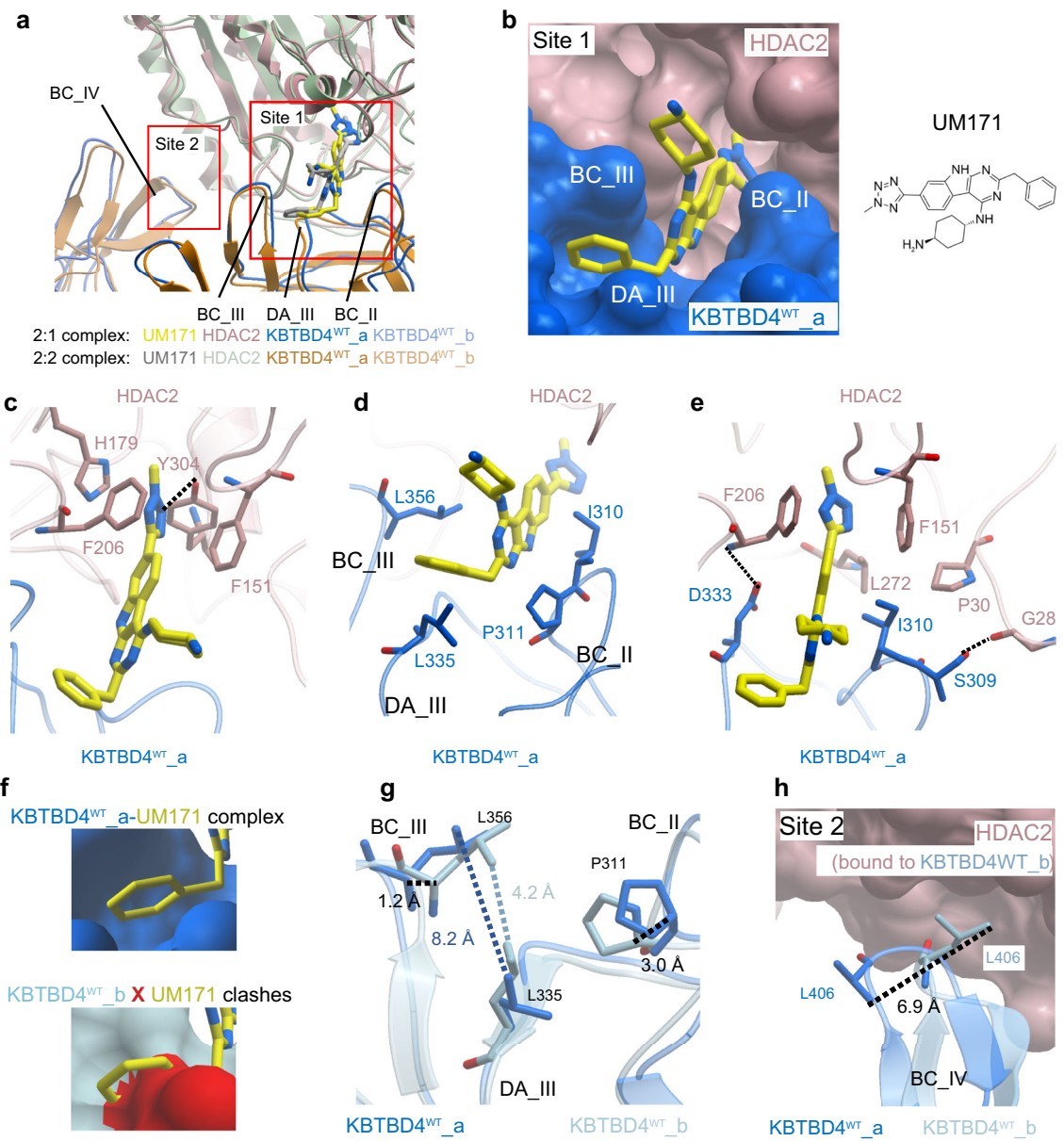

**Fig. 6 | Cryo-EM structures show UM171 recruits HDAC2 to KBTBD4^WT as a molecular glue. a** Superposition of 2:1 and 2:2 complexes at HDAC2-interacting sites 1 and 2 in ribbon representation. UM171 is shown in stick representation. KBTBD4^WT BC loops involved in both interaction sites are labelled. **b** Chemical structure of UM171 and a close-up view of UM171 binding at site 1 in the KBTBD4^WT-HDAC2 interface. UM171 is shown in yellow sticks, KBTBD4^WT and HDAC2 are shown in blue and pink as surface representations. **c–e** Close-up views of UM171 binding showing key interfacial residues as sticks. Hydrogen bonds are indicated by dashed lines. The subunits are coloured as in **b**. **f** Surface representations show the

induced-fit of UM171 bound to KBTBD4^WT_a (top panel) relative to the equivalent surface in unbound KBTBD4^WT_b (lower panel). Loops involved in the induced fit are labelled. **g** Superposition of UM171-bound KBTBD4^WT_a (blue) and unbound KBTBD4^WT_b (light blue) at site 1 in ribbon and stick representations. Distances are measured to show the enlarged cleft in the UM171-bound KBTBD4^WT_a. **h** Superposition of KBTBD4^WT site 2 in HDAC2-bound KBTBD4^WT_b and unbound KBTBD4^WT_a. KBTBD4^WT chains are shown in ribbon representation. L406 in BC IV in both KBTBD4^WT chains is shown in sticks and the distances between side chains is measured to demonstrate the loop movement.

Cα positions. Superposition of the 2:1 and 2:2 states shows that the high structural conservation extends across the protein-compound and protein-protein interaction sites (Fig. 6a), indicating that the neo-substrate recruitment mechanism in these complexes is conserved regardless of the stoichiometry. The additional binding of CoREST1, however, appears to enforce an asymmetric 2:1:1 assembly due to steric constraints, as noted above (Fig. 4d and Supplementary Fig. 5). Overall, the models reveal similar site 1 and site 2 binding interfaces to the mutant KBTBD4 complexes, with both Kelch domains in the KBTBD4^WT dimer contacting each subunit of HDAC2 (Fig. 6a). The CoREST1^NT2 protein wraps around HDAC2 similarly to other ELM2-SANT domain proteins[29].

UM171 is buried in the site 1 interfaces where it forms contacts with blades II and III from the KBTBD4^WT Kelch domain, as well as with the HDAC2 catalytic pocket. Importantly, the maps allowed full tracing of the UM171 scaffold (Fig. 6b and Supplementary Fig. 3). The methyl tetrazole inserts into the HDAC2 catalytic pocket to form extensive interactions, including a hydrogen bond with HDAC2 Y304 and π–π interactions with HDAC2 F151, F206 and H179 (Fig. 6c). The binding is further stabilised by π–π interactions between the buried UM171 pyr-imidoindole core and HDAC2 residues F151 and F206. Unlike SAHA and other HDAC inhibitors, UM171 does not coordinate the catalytic Zn^{2+} ion. Outside of the HDAC2 catalytic pocket, UM171 forms hydrophobic interactions with KBTBD4^WT I310 and P311 from the BC II loop and L356

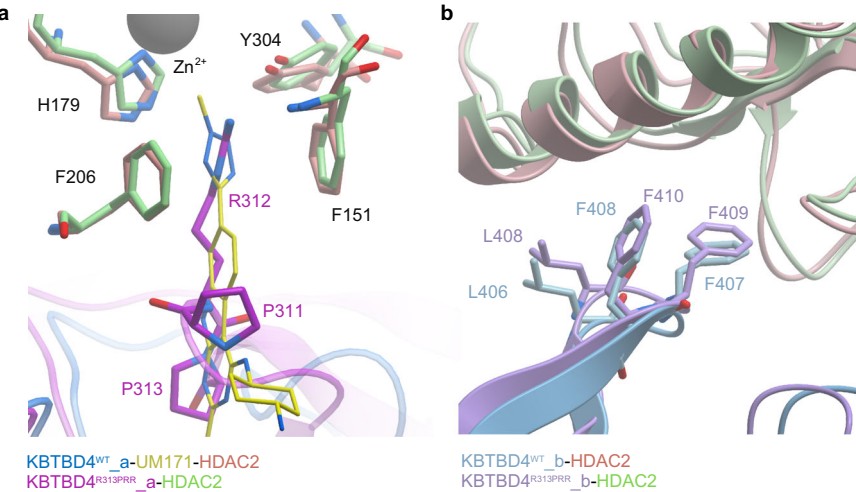

**Fig. 7 | Structural mimicry of UM171 and cancer mutant KBTBD4$^{R313PRR}$.** Superposition of UM171 and KBTBD4$^{R313PRR}$ at site 1 (**a**) or site 2 (**b**). UM171 and key interacting residues are shown as sticks. Subunits are coloured as following: in the UM171 model, KBTBD4$^{WT}$ chains in blue and light blue, UM171 in yellow and HDAC2 in pink; in the KBTBD4$^{R313PRR}$ model, KBTBD4$^{R313PRR}$ in purple and light purple and HDAC2 in green.

located at the tip of the BC III loop (Fig. 6d). Together L356 (BC III loop) and L335 (DA III loop) establish a narrow cleft for the UM171 benzyl group to anchor (Fig. 6d). Finally, direct interactions between KBTBD4$^{WT}$ S309, I310, D333 and HDAC2 G28, P30, F151, F206, L272 complete the ternary interaction network and highlight the native surface compatibility of the two proteins (Fig. 6e).

The asymmetric 2:1 and 2:1:1 models allowed comparison of the equivalent KBTBD4$^{WT}$ site 1 in both the bound and unbound states. Interestingly, UM171 was only observed in the HDAC2-bound site 1 and not in the second Kelch domain, suggesting that UM171 binding requires both proteins to be present. Indeed, UM171 did not exhibit pronounced affinity towards either KBTBD4$^{WT}$ or HDAC2 alone in an MST assay (Supplementary Fig. 9), highlighting the importance of the ternary interface for forming a suitable ligand-binding pocket. These features are consistent with previously defined molecular glue systems[48,49], establishing UM171 a bona fide molecular glue for KBTBD4$^{WT}$ and HDAC2. Notably, UM171 binding exhibited an induced-fit mechanism requiring the opening of a cleft between Kelch domain blades II and III to accommodate the benzyl group of UM171 (Fig. 6f). In the UM171-bound KBTBD4$^{WT}$ protomer, L356 (BC III loop) is shifted a further 3.4 Å away from P311 (BC II loop) and 4.1 Å from L335 (DA III loop) relative to the unbound site 1 in the other KBTBD4$^{WT}$ protomer (Fig. 6g). Such induced conformational change demonstrates a plasticity of the Kelch domain that hitherto was not widely observed.

Additionally, each HDAC2 subunit engages a second KBTBD4$^{WT}$ protomer at site 2 involving blade IV of the second Kelch domain, as observed in the mutant complex (Fig. 6h). Similar to site 1, site 2 in the 2:1 model shows subtle conformational changes between the HDAC2-bound and unbound states. In particular, KBTBD4$^{WT}$ L406 in the BC IV loop shows a shift of 6.9 Å (Fig. 6h) to pack together with KBTBD4$^{WT}$ F408 and F409 into a shallow hydrophobic pocket on the HDAC2 surface. The equivalent F408 and F409 side chains were not resolved in the second KBTBD4$^{WT}$ protomer, suggesting their flexible nature when not engaged in HDAC2 interaction. Taken together, the structural analyses show that ternary complex formation is primed by native surface complementarity and stabilised by UM171, which acts as a molecular glue to bridge across the site 1 interface.

### Structural mimicry of UM171 and cancer mutant KBTBD4$^{R313PRR}$
The overall assemblies of the KBTBD4$^{R313PRR}$-HDAC2 and KBTBD4$^{WT}$-UM171-HDAC2 complexes are highly conserved (Cα RMSD < 3 Å). Site 1 superposition reveals that UM171 mimics the mutated BC II loop of

KBTBD4$^{R313PRR}$ by engaging the same HDAC2 residues in its catalytic pocket (Fig. 7a). In particular, the methyl tetrazole of UM171 superimposes with the side chain of KBTBD4$^{R313PRR}$ R312 and forms similar interactions. Outside of the HDAC2 pocket, the pyrimidoindole core of UM171 overlays with the main chain atoms of the 311-PRP-313 loop of the KBTBD4$^{R313PRR}$ structure (Fig. 7a). Similarly, the binding pose adopted by KBTBD4$^{WT}$ blade IV at site 2 is conserved with its counterpart in the mutant complex despite the mutational frameshift in the preceding blade II (Fig. 7b). Such mimicry between a small molecule and natural mutation has been rarely reported and may inspire future strategies for molecular glue development utilising the knowledge of E3 ligase natural or engineered variants and their associated proteins.

### SAR of UM171 derivatives
In parallel, we carried out a systematic structure-activity relationship (SAR) analysis for UM171 revealing knowledge complementary to our structural study. We identified four sites (R1-R$^4$) in UM171 (**1**) for modifications. The R$^1$-R$^4$ substituents are positioned on the opposite sides of UM171 and represent 3 different vectors. A naïve assessment suggested that R$^1$ and R$^3$ might be in closer contact with the protein, while R$^4$ might be directed towards the solvent. To explore these sites, we designed compounds (**2-15**) (see the synthesis report in the Supplementary Information). The impact of the modifications was assessed in a cellular degradation assay for which HEK293 cells were incubated with these compounds at 1 μM for 16 hours. Past proteomic analyzes revealed greater degradation of CoREST and LSD1 than HDAC1/2[22–24]. Therefore, the cellular assay focused on the levels of CoREST1 and LSD1, which for each compound were normalised and calculated in relation to DMSO control, as summarized in Table 2 and Supplementary Fig. 10.

At the R$^1$ site, substitution of the methyl tetrazole (**1**) with a 3,4-dimethylisoxazole (**3**) led to the complete loss of activity. In contrast, a carboxylic acid methyl ester at R$^1$ (**4**) rescued the activity and caused a robust degradation of both LSD1 and CoREST1 even exceeding the activity of the reference compound UM171, suggesting smaller moieties at R$^1$ might be favoured. However, substituting the methyl ester in (**4**) with a carboxylic acid moiety (**5**), or adding a nitrile at R$^1$ (**6**) did not show detectable activity, potentially due to desolvation penalty or lack of apolar interactions. These observations validated the importance of the hydrophobic interactions observed in the cryo-EM structures between the UM171 methyl tetrazole and the HDAC2 catalytic pocket containing F151, H179, F206, L272 (Supplementary Fig. 11). However,

the activity loss by an isosteric methylpyrazole moiety replacement (**2**) is hard to explain by the current structural information and probably merits further chemical exploration.

Similar to R[1], changes at the R[2] and R[3] sites significantly impacted the compound activity. Substituent exchange between R[1] and R[2] was explored by compounds (**4**) and (**7**). Moving the methyl ester moiety to the adjacent position caused loss of activity, potentially due to a steric clash with the narrow HDAC2 catalytic pocket. For R[3], replacing the benzyl moiety with a hydrogen (**8**) reduced the activity, validating the importance of this benzyl moiety in engaging KBTBD4 as shown in the cryo-EM structures (Fig. 6 and Supplementary Fig. 11). Compound activity is sensitive to changes at these sites, potentially owing to their proximity to the buried KBTBD4-HDAC2 interface as shown in the structures.

In contrast, changes at R[4] were more tolerable, in line with our original hypothesis as well as the cryo-EM structures that this moiety is exposed to solvent. Substitution of the primary amine with dimethylamine (**9**) led to no perceptible change in CoREST1 or LSD1 levels, reinforcing that the activity changes observed for compounds (**4**), (**5**), and (**8**) were solely owing to the variability in R[1] and R[3] but not R[4]. Removing the distal basic moiety at R[4] (**10**) resulted in reduced activity, while incorporating a basic pyrrolidine (**11**) or piperidine (**12**) ring rescued the activity with a slightly better potency observed for compound (**11**). As the R[4] substituent is away from the complex binding interface in the cryo-EM structures, we speculate its observed impact on substrate degradation may be due to varied physico-chemical properties and cell permeability of the compound analogues. Interestingly, a carboxylic acid methyl ester at R[1] consistently showed superior activity to the methyl tetrazole moiety in the presence of three different R[4] substituents (comparing **4-9**, **11-13** and **12-14** compound pairs), suggesting the two sites function independently and can be optimised separately.

Finally, we examined the impact of the central ring of (**1**). Compound (**15**) with an open central ring exhibited no activity, suggesting that the relative orientation of the R[1]-R[4] substituents ensured by the UM171 pyrimidoindole core, is essential for the molecular glue effect.

Using the MST assay, we were able to observe a 3.5-fold improvement in the ternary complex affinity of the most active analogue (**13**) (S234984), whereas there was a 3-fold reduction for the inactive compound (**2**), confirming the expected correlation between ternary complex stability and cellular activity (Supplementary Fig. 12). Taken together, our experimental structural-activity relationship analysis, as well as the cryo-EM structures, extend our understanding of UM171 and will guide future rational optimisation for the drug.

## Discussion

Here, we elucidate the neomorphic mechanism of KBTBD4. Our cryo-EM and supporting data show how cancer-driving mutations remodel the KBTBD4 surface and identify HDAC1/2 as the direct interactor for recruitment of the LSD1-CoREST-HDAC1/2 complex. We found that UM171, a clinical phase 2 drug for haematopoietic stem cell transplantation, structurally mimics the cancer-mutated loop and acts as a molecular glue for wild-type KBTBD4 and HDAC1/2, resolving the long-standing puzzle of UM171's mechanism of action.

Our structures highlight several important features of the neo-substrate recruitment. Overall, the mutant and drug-induced neo-complexes adopt highly conserved structural assemblies. Consistent with current knowledge, the binding partners exhibit native surface complementarity that is enhanced by UM171 and the cancer mutant. Both Kelch domains in the KBTBD4 homodimer provide distinct epitopes for the HDAC2 subunit to bind, demonstrating the importance of the dimer for bivalent substrate recognition, as observed in our biochemical data. Contrary to expectation, these binding sites occur at the outer rim of the Kelch β-propeller domain rather than the central

pocket typically utilised by native Kelch E3 substrates[32,33,35–38]. UM171 acts at the ternary interface to remodel the Kelch domain surface similar to the cancer mutant. Our structure-activity relationship (SAR) analysis provides a basis for further rational design of UM171. Similar to IMiDs[50], parts of the UM171 scaffold display a steep SAR, likely reflecting the relatively high contribution (~40%) of the ligand to the complex interface. Nonetheless, we identify sites within UM171 amenable to chemical modification for improved potency guided by the 3D structural models.

In cells, HDAC1/2 can engage different ELM2-SANT domain proteins to assemble into distinct repressor complexes. In addition to CoREST, UM171 induces some degradation of MIER2, but seemingly not of the NuRD complex containing MTA1[24]. Notably, CoREST and MIER2 are degraded to a greater extent than the direct interactor HDAC1/2, suggesting that in the context of these large multi-subunit complexes HDAC1/2 is less accessible for ubiquitination. Our structures provide opportunity to model the different repressor complexes to gain potential insights into the neomorphic substrate specificity. Modelling of MTA1[51] predicts a steric clash between its BAH domain and KBTBD4 that may preclude its binding (Supplementary Fig. 5). Steric constraints also predict that the KBTBD4 dimer can accommodate only one HDAC1/2-CoREST pair, as observed in our 2:1:1 complex.

In support of our HDAC2-CoREST model, parallel but complementary approaches using HDAC1-CoREST have yielded mutant and UM171-dependent KBTBD4 complexes with a conserved 2:1:1 structural arrangement[52,53]. An interesting feature of these structures was the inclusion of inositol hexakisphosphate (InsP$_6$), a known HDAC1/2 cofactor. InsP$_6$ binding stabilised the HDAC1-CoREST interface as expected, but also formed a second molecular glue interaction with KBTBD4 to enhance the ternary complex affinity, as we have also observed in our MST assay (Supplementary Fig. 13).

To date, only a small number of E3s have been shown to engage a molecular glue. KBTBD4 expands this E3 repertoire to the CRL3 family and may offer a distinct interaction surface for target substrates less compatible with previously reported E3s. The exquisite mimicry between UM171 and KBTBD4[R313PRR] raises the intriguing prospect that mutations could help to identify other exploitable E3-target interfaces and inspire the discovery of further molecular glues. While only a few mechanistic studies exist[14,54], a recent proteome-wide screen of cancer mutants has revealed that neomorphic E3 interactions may be relatively prevalent[3]. Understanding these mechanisms will inform cancer biology, as well as help to define other E3-neo-substrate interfaces that may be hijacked by small molecules. Learning from UM171 and KBTBD4, future molecular glue discovery can potentially be accelerated by prioritising E3 ligases selected by nature, or by wider genetic screens.

## Methods

### Plasmid construction

FLAG-KBTBD4[WT] (17-534 a.a., Uniprot Q9NVX7-2) and FLAG-KBTBD4[R313PRR] (17-536 a.a., cancer indel mutation introduced by site-directed mutagenesis) were subcloned into a pcNDA3.1 (+) vector with the 3xFLAG epitope tagged to its N-terminus using standard restriction enzyme cloning[14]. The Kelch domain of KBTBD4[R313PRR] (237-536 a.a.) was amplified by PCR from the full-length mutant construct and subcloned into a pcDNA3.1(+) vector with the 3xFLAG epitope tagged to its N-terminus using standard restriction enzyme cloning. Monomeric KBTBD4[R313PRR] was generated by introducing mutations H42A, V46D, I50T, L53E, F60S, L77K and A81E and cloned into the pCMV-3Tag-1A vector using the custom gene synthesis service provided by GenScript.

For interaction site mapping in HEK293 cells, full-length human CoREST1 (pCMV5 HA RCOR1; NM_015156.4) construct was purchased from the MRC Protein Phosphorylation and Ubiquitylation Unit (cat#

**Table 2 | Structure-activity relationship of UM171 analogues in the degradation of CoREST1 and LSD1 in HEK 293 cells**

| No | R¹ | R² | R³ | R⁴ | LSD1 rel level* / % | CoREST1 rel level* / % |
|---|---|---|---|---|---|---|
| 1 (UM171) | | H | | | 28 ± 2 | 22 ± 5 |
| 2 | | H | | | >100 | >100 |
| 3 | | H | | | >100 | >100 |
| 4 | | H | | | 17 ± 6 | 5 ± 0.1 |
| 5 | | H | | | >100 | 95 ± 10 |
| 6 | | H | | | >100 | >100 |
| 7 | H | | | | 92 ± 25 | >100 |
| 8 | | H | H | | 41 ± 11 | 45 ± 18 |
| 9 | | H | | | 29 ± 4 | 21 ± 6 |
| 10 | | H | | | 98 ± 23 | 90 ± 24 |
| 11 | | H | | | 31 ± 3 | 14 ± 6 |
| 12 | | H | | | 46 ± 12 | 36 ± 9 |
| 13 (S234984) | | H | | | 15 ± 5 | 5 ± 0.3 |
| 14 | | H | | | 27 ± 9 | 4 ± 1 |
| 15 | N/A | N/A | N/A | N/A | >100 | >100 |

*The immunoblots of corresponding proteins were quantified and normalised to the GAPDH loading controls. The relative protein levels were calculated in relation to the DMSO controls on the same blot (n = 3; errors represent S.E.M.). Source data are provided in the Source Data file.

DU19711, https://mrcppureagents.dundee.ac.uk). N or C-terminal fragments of CoREST1 were generated using standard PCR methods, followed by restriction enzyme cloning into a pCMV5 vector with an N-terminal HA fusion.

For biophysical assays and cryo-EM studies, the ELM2-SANT1 domain region of human CoREST1 (71-259 a.a., Uniprot Q9UKL0-1) was amplified from the full-length construct above and cloned into the mammalian expression vector pHTBV with a C-terminal FLAG/10x-histidine fusion using ligation-independent cloning. The untagged

full-length HDAC2 (1-488 a.a., Uniprot Q92769-1) was cloned into the pcDNA3.1(−) vector using standard restriction enzyme cloning. For microscale thermophoresis, near full-length LSD1 (171-852 a.a., Uniprot O60341-1) was cloned using ligation-independent cloning into bacterial expression vector pNIC28-Bsa4 (GenBank accession number EF198106) which provides an N-terminal 6x-histidine tag and a TEV cleavage site.

For ubiquitination assays, full-length CUL3 (1-768 a.a., Uniprot Q13618-1) was cloned into a baculovirus transfer vector pFAST-Bac™

(Invitrogen) with a C-terminal FLAG/6x-histidine tag and a TEV protease cleavage site using ligation-independent cloning. Full-length RBX1 was cloned into pFAST-Bac™ with no tag. The neddylation enzyme plasmids were gifts from Brenda Schulman's lab and were described in their previous publication[55]. The ubiquitin E1 and E2 plasmids were gifts from Cheryl Arrowsmith (Addgene plasmids #25213, #25598, #25145, #25470). All DNA sequences were verified by Source Bioscience Ltd.

DNA oligonucleotides used for plasmid construction are listed below:

| | | |
|---|---|---|
| Full length FLAG-KBTBD4$^{WT}$ | fwd | CCGGCCGGATCCATGGAATCACCAGAGGAGCCTGG |
| | rev | GCATACGTCGACTTAGGCCAACACAAACTGCAAATTG |
| Full length FLAG-KBTBD4$^{R313PRR}$ mutagenesis | fwd | TTGTATGGTGGGAGGGTCCATCCCACGGCCACGGCGCATGTGGAAGT |
| | rev | ACTTCCACATGCGCCGTGGCCGTGGGATGGACCCTCCCACCACATACAA |
| Kelch domain of KBTBD4$^{R313PRR}$ (237-536 a.a.) | fwd | CCGGCCGGATCCATGGAGGCTTTTGCAGAGTCACTCAGGA |
| | rev | GCATACGTCGACTTAGGCCAACACAAACTGCAAATTG |
| CoREST1-NT1 | fwd | GCATACGGATCCATGCCGGCCATG |
| | rev | CCGGCCGTCGACTCACTCCTCCCGCTCCCG |
| CoREST1-NT2 | fwd | GCATACGGATCCATGAAAAGTTTGGCGGCGGC |
| | rev | CCGGCCGTCGACTCACTCCTCCCGCTCCCG |
| CoREST1-CT1 | fwd | GCATACGGATCCATGAGCGAGGATGAACTGGAAGAGG |
| | rev | CCGGCCGTCGACTCAGGAGGCAGATGCATATCTGAC |
| CoREST1-CT2 | fwd | GCATACGGATCCATGGTCAAAAAAGAAAAACATAGCACACAAGCTAAAAATAG |
| | rev | CCGGCCGTCGACTCAGGAGGCAGATGCATATCTGAC |
| CoREST1 (71-259 a.a) for cryo-EM | fwd | TTAAGAAGGAGATATACTATGTCCTGGGAGGAAGGCAGC |
| | rev | GATTGGAAGTAGAGGTTCTCTGCGCGATCCATCACACTAGTT |
| Full length HDAC2 (1-488 a.a.) for cryo-EM | fwd | TTAAGAAGGAGATATACTATGTCCTGGGAGGAAGGCAGC |
| | rev | GATTGGAAGTAGAGGTTCTCTGCGCGATCCATCACACTAGTT |
| Near full-length LSD1 (171-852 a.a.) | fwd | TTAAGAAGGAGATATACTATGCCATCGGGTGTGGAGGGC |
| | rev | GATTGGAAGTAGAGGTTCTCTGCTCACATGCTTGGGGACTGCTG |
| CUL3 (1-768 a.a.) | fwd | GATTGGAAGTAGAGGTTCTCTGCTCACATGCTTGGGGACTGCTG |
| | rev | |
| | | GATTGGAAGTAGAGGTTCTCTGCTGCTACATATGTGTATACTTTGCGATC |
| RBX1 (1-108 a.a.) | fwd | TACTTCCAATCCATGGCGGCAGCGATGGATGTGG |
| | rev | TATCCACCTTTACTGTCAGTGCCCATACTTTTGGAATTCCC |

## Immunoprecipitation and immunoblotting

HEK293 cells (ATCC CRL-1573) were cultured in high glucose Dulbecco's Modified Eagle's Medium (GIBCO #41965039) with 5% penicillin-streptomycin (ThermoFisher Scientific #15070063) and 10% fetal bovine serum (Sigma-Aldrich #F9665) inside a 5% $CO_2$ incubator at 37 °C. Cells were transfected as indicated at 60% confluency (5 µg DNA per plate using PEI-MAX 40 kDA at mass ratio 3:1). After 40 h, cells were treated with 0.1 µM MLN4924 (Calbiochem, #951950–33–7) for 4 h. Cells were washed with phosphate-buffered saline, harvested into lysis buffer (50 mM Tris pH 7.5, 150 mM NaCl, 10% glycerol, 0.1% NP-040, 1 mM EDTA and 5 mM $MgCl_2$) containing protease inhibitors (Sigma-Aldrich #11697498001), beta-glycerolphosphate, DTT, PMSF and okadaic acid, incubated on ice for 30 min and centrifuged for 30 min at 20 000 g at 4 °C. The supernatant was incubated with Anti-FLAG M2 affinity gel (Sigma-Aldrich #A2220) or HA affinity gel (Sigma-Aldrich, #E6779) at 4 °C for 3 h before being washed 3 times with lysis buffer. Immunoprecipitates were eluted using 1× LDS buffer (ThermoFisher Scientific #NP0007) supplemented with DTT and boiled for 3 min at 95 °C. Cell lysates or immunoprecipitates were resolved in 4-12% gradient Bis-Tris gels, transferred to PVDF membranes (Merck, #IPVH00010) and immunoblotted with corresponding antibodies at 1:1000 dilution – anti-FLAG (Sigma-Aldrich #F1804), anti-HA (Roche, #12013819001), anti-RCOR1 (Cell Signalling Technology, #14567), anti-LSD1 (Cell Signalling Technology, #2139) and anti-HDAC2 (Cell Signalling Technology, #5113). Data were acquired and analyzed in Image Lab 5.2.1. Source data are provided in the Source Data file.

## Protein expression

5 mg DNA of KBTBD4 alone or CoREST1$^{NT2}$ and HDAC2 constructs at 1:1 ratio was transfected into 1 L Expi293F™ (Thermo Fisher Scientific #A14528) cell culture in mid-log phase ($2 \times 10^6$ mL$^{-1}$) in Freestyle 293™ Expression Medium (Thermo Fisher Scientific #12338018) with 15 mg PEI-MAX 40 kDA. Cells were grown in an orbital shaker at 37 °C with 8% $CO_2$ for 24 h and then 30 °C for 48 h before being harvested by centrifugation at 800 g for 12 min. The pelleted cells were washed with phosphate-buffered saline, pelleted again, then flash-frozen in liquid nitrogen for storage in a −80 °C freezer.

All bacterially expressed plasmids were transformed into E. coli strain BL21(DE3)R3-pRARE2. Cells were cultured in LB broth at 37 °C until OD$_{600}$ reached 0.6. Recombinant protein expression was then induced by addition of 0.4 mM isopropyl β-D-1-thiogalactopyranoside, followed by 18 h continuous shaking at 18 °C. Cells were harvested by centrifugation and flash-frozen in liquid nitrogen for storage in a −80 °C freezer.

For baculoviral expression, recombinant baculovirus was generated by site-specific transposition in the DH10Bac™ E. coli strain (Thermo fisher, #10361012), followed by bacmid isolation and transfection into Sf9 insect cells using JetPrime® (Polyplus, 114-15). Amplified P2 virus was used to infect Sf9 insect cells at $2 \times 10^6$ mL$^{-1}$. Infected cells were incubated at 27 °C with 100 rotations per minute (RPM) shaking for 48 to 72 hours before being harvested by centrifugation at 800 g in 4 °C for 12 min. The pelleted cells were washed with phosphate-buffered saline, pelleted again, then flash-frozen in liquid nitrogen for storage in a −80 °C freezer.

## Protein purification

Whole mammalian cell pellets expressing KBTBD4 or CoREST1[NT2]-HDAC2 complex were resuspended in buffer A (50 mM 4-(2-Hydroxyethyl)piperazine-1-ethane-sulphonic acid (HEPES) pH 7.5, 150 mM NaCl, 10% glycerol, 5 mM MgCl$_2$, 0.5 mM TCEP) supplemented with protease inhibitors and lysed by sonication. Recombinant proteins were captured by anti-FLAG M2 affinity gel, washed with buffer A supplemented with 10 mM ATP and 20 mM MgCl$_2$, and eluted with 3xFLAG peptide (ThermoFisher #A36806). The sample was then concentrated and subjected to size-exclusion chromatography using a Superdex S200 10/300 GL increase column pre-equilibrated with buffer A. Peak fractions were pooled and concentrated to 20 µM.

For all bacterial and insect cell expressed proteins, cell pellets were resuspended in buffer B (50 mM HEPES pH 7.5, 500 mM NaCl, 5% glycerol, 5 mM imidazole, 0.5 mM TCEP). LSD1 protein was captured on nickel sepharose resin, washed with buffer B and eluted by a stepwise gradient of 30–250 mM imidazole. Further clean-up was performed by size-exclusion chromatography using a HiLoad Superdex S200 16/60 or S75 column buffered in 50 mM HEPES pH 7.5, 300 mM NaCl, 0.5 mM TCEP. Peak fractions were pooled and concentrated to 20 µM.

## Microscale thermophoresis

Experiments were performed in a Monolith NT.115 instrument (NanoTemper Technologies GmbH). Purified recombinant KBTBD4 variants, as well as HDAC2 were labelled using a Monolith RED-NHS 2[nd] generation Protein Labelling Kit following the manufacturer's instruction. CoREST1[NT2]-HDAC2, HDAC2, UM171, compounds (**2**), (**13**) or BRD6929 (MedChemExpress #HY-100719) as indicated were mixed in each experiment with 40 nM labelled KBTBD4 or 80 nM HDAC2 buffered in 50 mM HEPES pH 7.5, 150 mM NaCl, 10% glycerol, 5 mM MgCl$_2$, 0.05% Tween-20, 0.1 mM TCEP, with or without 40 µM inositol hexaphosphate. The reactions were incubated at room temperature for 15 min and then transferred into the standard MST capillaries for measurements at 25 °C with 20% excitation power and 20% or 40% MST power in MO. Control 1.6.1. Data were analyzed using the Monolith Evaluation software MO. Affinity Analysis 2.3 using the implemented $K_D$ model and plotted in GraphPad Prism 10.4.0. Source data are provided in the Source Data file.

## Crosslinking assay

Purified proteins at equal molar ratio were mixed as indicated in 50 mM HEPES pH 7.5, 150 mM NaCl, 10% glycerol, 5 mM MgCl$_2$, 0.1 mM TCEP and incubated at room temperature for 15 min. Bissulfosuccinimidyl Suberate (BS$^3$, ThermoFisher Scientific #21580) was then added to reach a final concentration of 4 mM. Reactions were incubated at room temperature for 45 min and then terminated by adding either 1x LDS buffer or 50 mM Tris pH 7.5.

## In vitro ubiquitylation

4 µM purified full-length CUL3-RBX1 was mixed with 0.5 µM APPBP1-Uba3, 1 µM UBE2M and 60 µM NEDD8 in 50 mM HEPES pH 7.5, 150 mM NaCl, 10% glycerol, 0.5 mM TCEP, 10 mM MgCl$_2$ and 10 mM ATP for 30 min at room temperature for in vitro neddylation[55]. 100 µM ubiquitin, 100 nM UBA1, 1 µM neddylated CUL3-RBX1, 1 µM UBE2 of choice and 5 µM CoREST1[NT2]-HDAC2 were mixed in 50 mM HEPES pH 7.5, 150 mM NaCl, 10% glycerol, 0.5 mM TCEP, 10 mM MgCl$_2$ and 10 mM ATP. 1 µM KBTBD4 (WT or R313PRR) and 10 µM UM171 were added into the reactions as indicated. Reactions were incubated at room temperature for 30 min, subjected to SDS-PAGE, and then analyzed by Coomassie staining or immunoblots.

## Cryo-EM sample preparation, data acquisition and data processing

For the KBTBD4[WT-UM171]-CoREST1[NT2]-HDAC2 complex, 4 µM KBTBD4[WT], 4 µM CoREST1[NT2]-HDAC2 and 40 µM UM171 were mixed in 50 mM HEPES pH 7.5, 150 mM NaCl, 5% glycerol, 5 mM MgCl$_2$, 0.5 mM TCEP, and then crosslinked with BS$^3$. Samples were plunge-frozen on Quantifoil Cu R1.2/1.3 300-mesh grids that had been freshly glow-discharged for 1 min. The Vitrobot for plunge-freezing was set to 100% humidity, 4 °C and 4 s blotting time.

For the KBTBD4[R313PRR]-CoREST1[NT2]-HDAC2 complex, 4 µM KBTBD4[R313PRR] and 4 µM CoREST1[NT2]-HDAC2 were mixed in 50 mM HEPES pH 7.5, 150 mM NaCl, 10% glycerol, 5 mM MgCl$_2$, 0.5 mM TCEP, and incubated for 15 min on ice. Samples were plunge frozen on UltrAufoil R1.2/1.3 300-mesh grids which had been freshly glow-discharged for 1 min. The Vitrobot Mark IV (ThermoFisher Scientific) for plunge-freezing was set to 100% humidity, 4 °C and 2.5 s blotting time. All grids were clipped into autogrid cartridges (Thermo Fisher #1036173).

Both datasets were collected on a FEI Titan Krios (Thermo Fisher Scientific) operating at 300 kV at the Central Oxford Structural Molecular Imaging Centre (COSMIC). Super-resolution micrographs with 2x binning (0.832 Å pixel⁻¹) were collected at ×105,000 nominal magnification at 38 or 38.6 e⁻/Å² dose using a Gatan K3 direct electron detector.

For the KBTBD4[WT] dataset, 12,722 micrographs with defocus range −3.2 to −1.0 µm were collected and then processed in Cryosparc v4.4.1. 13,588,054 particles were picked with the blob picker function, and extracted particles were subjected to multiple cycles of 2D classification. 193,056 particles of good classes were selected to generate an ab-initio model which revealed partial occupancy for a segment suggesting heterogeneity. Particles were then manually separated based on 2D classes to generate ab-initio models, and then subjected to a heterogenous refinement which revealed distinct 2:1 and 2:2 species. The preliminary 3D volumes were used to create 2D templates for rare view particle picking with the template picker function. The new particles were subjected to 2D classifications, resulting in 264,324 and 356,933 particles of good classes used in non-uniform refinements for 2:1 (C1 symmetry) and 2:2 (C2 symmetry), respectively. 2:1 and 2:2 maps were further refined to 3.1 and 2.9 Å. Additionally, a subclass of 2:1 species containing a CoREST1 subunit was identified by a 3D variability job and further refined to 3.3 Å with 105,484 particles by local refinement.

For the KBTBD4[R313PRR] dataset, 13,131 micrographs with defocus range -2.8 to -1.0 µm were collected and processed similarly to the WT dataset. 295,217 particles were selected for 3D reconstitution after blob picking, template picking and multiple rounds of 2D classification. Non-uniform refinement using these particles with C2 symmetry applied reached a final resolution of 2.7 Å.

## Model building and refinement

The full-length KBTBD4 model generated by AlphaFold 2[56], deacetylase domain of HDAC2 (7ZZT) and ELM2-SANT1 domain of a CoREST1 analogue MTA1 (4BKX) were fitted into the reconstituted EM maps using UCSF ChimeraX 1.7.1[57]. The models were manually refined in Coot 0.8.9.2[58] and then auto-refined in Phenix 1.20.1-4487[59] using the strategies implementing real space, rigid body and atomic displacement parameters. The geometry of all the models were validated using Phenix_MolProbity.

## Compound synthesis and quality controls

A detailed synthesis report can be found in the Supplementary information. All reagents obtained from commercial sources were used without further purification. Anhydrous solvents were obtained from commercial sources and used without further drying. The reactions were monitored using LC-MS and GC-MS instruments.

Analytical LC-MS: Agilent HP1200 LC with Agilent 6140 quadrupole MS, operating in positive or negative ion electrospray ionisation mode. Molecular weight scan range was 100 to 1350 m/z. Parallel UV detection was done at 210 nm and 254 nm. Samples were supplied as a

1 mM solution in MeCN or in THF/water (1:1) with 5 μL loop injection. LC-MS analyses were performed on two instruments, one of which was operated with basic, and the other with acidic eluents.

Basic LC-MS: Gemini-NX, 3 μm, C18, 50 mm × 3.00 mm i.d. column at 23 °C, at a flow rate of 1 mL min-1 using 5 mM aq. NH4HCO3 solution and MeCN as eluents.

Acidic LC-MS: ZORBAX Eclipse XDB-C18, 1.8 μm, 50 mm × 4.6 mm i.d. column at 40 °C, at a flow rate of 1 mL min-1 using water and MeCN as eluents, both containing 0.02 V/V% formic acid.

Combination gas chromatography and low-resolution mass spectrometry were performed on Agilent 6850 gas chromatograph and Agilent 5975 C mass spectrometer using 15 m × 0.25 mm column with 0.25 μm HP-5MS coating and helium as carrier gas. Ion source: EI + , 70 eV, 230 °C, quadrupole: 150 °C, interface: 300 °C.

Flash chromatography was performed on ISCO CombiFlash Rf 200i with pre-packed silica-gel cartridges (RediSep®Rf Gold High Performance).

Preparative HPLC purifications were performed on an Armen Spot Liquid Chromatography system with a Gemini-NX® 10 μm C18, 250 mm × 50 mm i.d. column running at a flow rate of 118 mL min-1 with UV diode array detection (210 – 400 nm).

1H NMR and proton-decoupled 13 C NMR measurements were performed on Bruker Avance III 500 MHz spectrometer and Bruker Avance III 400 MHz spectrometer, using DMSO-d6 or CDCl3 as solvent. 1H and 13 C NMR data are in the form of delta values, given in part per million (ppm), using the residual peak of the solvent as internal standard (DMSO-d6: 2.50 ppm (1H) / 39.5 ppm (13 C); CDCl3: 7.26 ppm (1H) / 77.0 ppm (13 C)). Splitting patterns are designated as: s (singlet), d (doublet), t (triplet), q (quartet), sp (septet), m (multiplet), br s (broad singlet), dd (doublet of doublets), td (triplet of doublets), qd (quartet of doublets). In some cases, two sets of signals appear in the spectra due to hindered rotation.

HRMS were determined on a Shimadzu IT-TOF, ion source temperature 200 °C, ESI +/−, ionisation voltage: ( ± )4.5 kV. Mass resolution min. 10000.

All obtained products had an LC purity above 96% that was corroborated by their 1H NMR spectrum unless specifically mentioned otherwise.

### Reporting summary
Further information on research design is available in the Nature Portfolio Reporting Summary linked to this article.

## Data availability
Cryo-EM maps and atomic coordinates have been deposited in the Electron Microscopy Data Bank (EMDB) and Protein Data Bank (PDB) under accession codes EMD-51337 and PDB 9GGM (2:2 KBTBD4[R313PRR]-HDAC2), EMD-51338 and PDB 9GGN (2:2 KBTBD4[WT]-HDAC2 with UM171), EMD-51335 and PDB 9GGL (2:1 KBTBD4[WT]-HDAC2 with UM171) as well as EMD-51336 and PDB 9I2C<(2:1:1 KBTBD4[WT]-HDAC2-CoREST1 with UM171). All other data are available in the manuscript or the supplementary materials. Further information and requests for resources and reagents should be directed to and will be fulfilled by the corresponding author. Source data are provided with this paper.

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

## Acknowledgements

Cryo-EM studies were performed at the UK national Electron Bio-Imaging Centre (eBIC, proposal BI34631), the Central Oxford Structural Molecular Imaging Centre (COSMIC, supported by the Wellcome Trust grant no. 201536, the EPA Cephalosporin Trust and a Royal Society/Wolfson Foundation Laboratory Refurbishment grant no. WL160052), and the Oxford Particle Imaging Centre (OPIC, an Instruct-ERIC centre funded by Wellcome Trust JIF award 060208/Z/00/Z and equipment grant 093305/Z/10/Z). ZC, SCC, VD'A and ANB acknowledge funding from a Cancer Research UK grant DRCNPG-May21\100002. This study was also possible thanks to funding from the Innovative Medicines Initiative 2 Joint Undertaking (JU) under grant agreement number 875510 (EUbOPEN). JU receives support from the European Union's Horizon 2020 research and innovation programme and EFPIA and Ontario Institute for Cancer Research, Royal Institution for the Advancement of Learning McGill University, Kungliga Tekniska Hoegskolan, and Diamond Light Source. The scientific work/research and/or results publicised in this article were reached with the sponsorship of the Gedeon Richter Talentum Foundation in the framework of the Gedeon Richter Excellence PhD Scholarship of Gedeon Richter. We thank Brenda Schulman, Cheryl Arrowsmith and Melissa Sweeney for help with constructs.

## Author contributions

S.C.C., V.D.'A., An.K. and A.N.B. conceived the project. Z.C., X.C. and B.R.M. undertook protein production. Cryo-EM studies were performed by Z.C. and G.C. MST experiments were performed by Z.C. and X.C. UM171 derivatives were designed and/or synthesized by T.B., T.S., Ar.K., T.N., A.H. and An.K. Cellular studies for UM171 SAR were performed by T.B. and Z.C. and analyzed by T.B., Z.C., T.S., A.H., Ar.K. and An.K. Other

biochemical and cellular studies were performed by Z.C. Z.C. and G.C. prepared the figures. Z.C. wrote the initial manuscript with G.C., T.B., An.K. and A.N.B. All authors discussed and edited the manuscript.

## Competing interests

The authors declare no competing interests.
