## [Transparent Peer Review file · Nature Communications]

Structural mimicry of UM171 and neomorphic cancer mutants co-opts E3 ligase KBTBD4 for HDAC1/2 recruitment

Corresponding Author: Professor Alex Bullock

Version 0:

Reviewer comments:

Reviewer #1

(Remarks to the Author)

The manuscript by Chen et al. provides a characterization and structural analysis of the neomorphic KBTBD4 mutant (R313PRR) E3 ligase in complex with CoREST. Additionally, they demonstrated that UM171, a small molecule used in haematopoietic stem cell transplantation, functions as a molecule bridge between the wild type KBTBD4 and CoREST complex and further solved the structure. Structural comparisons reveal a mimicry in structure and similar binding modes between UM171 and the mutant KBTBD4 within the ternary complex. Furthermore, the authors performed SAR studies on UM171 and evaluated its efficacy in degrading CoREST, providing insight into the molecular mechanisms of the neomorphic KBTBD4 E3 ligase and could help future rational optimization for UM171. Overall, the manuscript appears to be appropriate interpretation on the data and the work will be of general interest.

I have the following comments:

Cryo-EM:

1. Regarding the model of WT KBTBD4: UM171-neosubstrate complex, the clash score of 20.12 is concerningly high. The model requires improvement and the extensive clashes need to be fixed.
2. There appears to be a discrepancy between the data collection parameters mentioned in the Methods and those listed in the Table 1. Were the data collected in super-resolution mode or counting mode? The defocus range does not align. Please verify these details.
3. Regarding the EM reconstruction for KBTBD4:HDAC2 (2:2) in supplementary Figure 1, the map exhibits orientation issues and poor quality. The authors should consider to include 3DFSC or similar metric to describe the anisotropy in data.
4. Cryo-EM maps in the figures: please specify what threshold the maps are displayed in the figure legends.
5. Validation of cryo-EM structural models (model-to-map FSC) should be provided.
6. In terms of deposition, it is recommended that the model for KBTBD4:HDAC2:CoREST1 should be also deposited.

Assays:

7. The cryo-EM structure showed that each HDAC2 molecule interacts with both Kelch domains within the KBTBD4 dimer, while the CoREST1 density is not well resolved, indicating a predominant role for HDAC2 in the CoREST complex binding to KBTBD4. Therefore, it makes sense that CoREST N terminal domain was mapped in Fig 2 as it contains the HDAC2 binding site. From this perspective, it seems that the interaction between KBTBD4 and HDAC2 is independent of CoREST1. What are the cross-linking results when the authors mix KBTBD4R313PRR with HDAC2 alone or CoREST1 alone? Is the affinity between KBTBD4R313PRR and HDAC2 equal to HDAC2-CoREST1? Does CoREST1 have any additional role in complex assembly? In Fig1, please also include IP for HDAC.
8. In Fig3, HDAC2 is shown to be ubiquitinated in KBTBD4R313PRR or KBTBD4WT-UM171 dependent manner, however, the degradation of the other two subunits, CoREST and LSD1 is much greater than that of HDAC1/2, and therefore used in SAR related degradation assay. Could the author explain this observation?
9. What are the affinity values obtained from the individual MST experiment depicted in Fig2c and Fig3b-c?
10. Have any mutagenesis, in vitro pull down or in vivo IP experiment been done to validate the structural details?

Minors:

- 1) In the Method section, "Recombinant protein expression was then induced by addition of 0.4 M isopropyl β -D-1-

thiogalactopyranoside", 0.4M or 0.4 mM?

2) The label for Raw blots for Figures 2f should be for Figure 3e.

3) For the raw blots, the labels for protein markers should be included and uncropped version should be provided.

Reviewer #2

(Remarks to the Author)

This is an interesting study that describes and compares the interactions of cancer mutant KBTBD4 with HDAC2 and the UM171-induced complex with HDAC2.

Overall, the manuscript has considerable strengths. It is a well-executed study with some genuinely important findings that make it suitable for publication in Nature Communications. Especially important is the mimicry between the cancer mutations and the effect of the small molecule UM171. I do however have two (related) major concerns that I believe can be addressed without further experiments, but by modifications to both the results and discussion sections.

1. Questionable relevance of the 2:2 structure given that CoREST is not bound to the HDAC2.

Given that the interaction with KBTBD4 is reported to be specific for the CoREST:HDAC complex and yet CoREST is not seen in the complex with mutant KBTBD4, it seems unlikely that the observed 2:2 complex is directly biologically relevant. It is possible that binding to KBTBD4 actively dissociates HDAC2 from the CoREST complex, but that does not fit with the CoREST & LSD1 pull-downs observed here and reported by others.

This does not necessarily mean that the structure is not biologically informative, but this issue needs to be addressed in some detail.

2. Specificity for the CoREST complex

The interaction and subsequent degradation of the CoREST:HDAC complex is reported as being rather specific. However the structures do not explain this specificity and this seems to be a major short-coming.

Based on analogy with the HDAC1:MTA1 and HDAC1:MiDAC complexes, and also alphafold predictions of the CoREST:HDAC2 complex, it should be easily possible to predict the disposition of the CoREST proteins with respect to HDAC2. Are these models compatible with the interaction observed for HDAC2 with KBTBD4? Is it possible to explain the specificity for the CoREST:HDAC complex? Or is the HDAC somehow extracted from the complex due to specific properties of the CoREST:HDAC:LSD1 complex?

Reviewer #3

(Remarks to the Author)

The authors present a compelling elucidation of UM171's mechanism of action, including structural insights into the UM171-induced interaction between KBTBD4 and HDAC2. Additionally, they demonstrate a striking structural mimicry between UM171 and a cancer-associated insertion in KBTBD4, as demonstrated by the parallel structural studies of the mutant complex. Additionally, some SAR data of the (now phase II) UM171 compound is presented. Overall, this study is well-executed and clearly described, and in my view it will be of high interest to the journal's broad readership. I am happy to recommend its publication, subject to consideration of the following - mostly minor - suggestions:

1. The authors do not reference a recent bioRxiv preprint from the Liao/Zheng labs that explores the same system and reaches similar (although subtly differing) conclusions. While I do not believe that this preprint in any way precludes publication, it would be prudent to acknowledge this parallel work within the manuscript, especially that largely complementary methodologies were used in the two studies.

2. Background interaction between KBTBD4 and HDAC2: The authors mention pre-existing complementarity and present data suggesting potential baseline binding (Fig. 3b, perhaps also 3a) and ubiquitination (Fig. 3e) in the absence of UM171, despite describing some of these as "background noise." Could the authors clarify whether this binding or ubiquitination is significant? E.g. can background interactions or ubiquitination be inhibited with SAHA? If so, is there observable turnover of the complex involving WT KBTBD4 in the baseline state?

3. Role of InsP6: previous studies have suggested a role for InsP6 in HDAC1/2 complexes. Furthermore, the other study proposed that the metabolite, in addition to UM171, is required for ternary complex formation - yet in this study, the complex is observed solely with UM171. Could the authors assess, perhaps via MST or another appropriate method, if InsP6 enhances binding synergistically?

4. C2 symmetry with HDAC2 vs asymmetric complex with HDAC1 in the Liao/Zheng lab preprint: Could the authors elaborate on their rationale for processing the structure with C2 symmetry and perhaps comment on the discrepancy between the two studies in how the HDAC1/2 is engaged?

5. For ease of interpretation, it would be beneficial to include the chemical structure of UM171 in Fig 6 (perhaps marking the R groups for the SAR study and this is the figure readers will refer to when reading Table 2)

6. SAR study: I suggest retesting 2-3 compounds using MST to distinguish effects related to physicochemical properties or cell permeability from effects on ternary complex formation. For the most promising compound, it would be helpful to check for binary interactions with each partner. Also, the observed difference between compounds 1 and 2 is notable—perhaps a retest in an in vitro setup is warranted?

7. Figures:

- a. Fig1a the labels are too close together, which might obscure interpretation
- b. Fig 5 panels d-f and h-j: these panels could be framed or re-arranged for clarity
- c. Fig. 5-7 or Supplementary Figures: Consider including a comparison of HDAC2 binding an acetyllysine of H3/4 to illustrate the mimicry point
- d. d. Some figures could be condensed, though this is a matter of preference

8. Text:

- a. The opening sentence of the abstract could benefit from a more positive framing
- b. In the final discussion paragraph, the first two sentences appear to conflate “target” and “ligase” – consider rephrasing to improve readability

Version 1:

Reviewer comments:

Reviewer #1

(Remarks to the Author)

The authors have addressed the concerns I raised in the previous review. I have one suggestion: all the replicated data should be included in the supplementary file to ensure comprehensive representation.

Reviewer #2

(Remarks to the Author)

I am happy that the authors have addressed my concerns and those of the other reviewers. I recommend publication without further revision.

Reviewer #3

(Remarks to the Author)

The authors have addressed the points raised in my initial review through both additional experiments and revisions to the text and figures. The study remains a well-executed and clearly presented contribution, and I appreciate the effort put into refining it further.

I am happy to recommend its publication in Nature Communications.

Response to Reviewers

We thank the reviewers for their helpful comments and suggestions. We include a point by point response addressing all of their comments below.

Reviewer #1 (Remarks to the Author)

The manuscript by Chen et al. provides a characterization and structural analysis of the neomorphic KBTBD4 mutant (R313PRR) E3 ligase in complex with CoREST. Additionally, they demonstrated that UM171, a small molecule used in haematopoietic stem cell transplantation, functions as a molecule bridge between the wild type KBTBD4 and CoREST complex and further solved the structure. Structural comparisons reveal a mimicry in structure and similar binding modes between UM171 and the mutant KBTBD4 within the ternary complex. Furthermore, the authors performed SAR studies on UM171 and evaluated its efficacy in degrading CoREST, providing insight into the molecular mechanisms of the neomorphic KBTBD4 E3 ligase and could help future rational optimization for UM171. Overall, the manuscript appears to be appropriate interpretation on the data and the work will be of general interest.

I have the following comments:

Cryo-EM:

1. Regarding the model of WT KBTBD4: UM171-neosubstrate complex, the clash score of 20.12 is concerningly high. The model requires improvement and the extensive clashes need to be fixed.

We have improved the model and reduced the clash score to 5.19. We have updated our PDB entry and uploaded the revised PDB file and validation report for review.

2. There appears to be a discrepancy between the data collection parameters mentioned in the Methods and those listed in the Table 1. Were the data collected in super-resolution mode or counting mode? The defocus range does not align. Please verify these details.

The Table 1 number of 12722 micrographs for the KBTBD4^{WT} dataset was correct, while the 12732 micrographs mentioned in the method section was a typo. We have corrected this. The data were collected in counting mode, which is practically super-resolution mode with 2x binning when using a K3 detector. The misaligned defocus ranges have now been corrected – this oversight was due to various defocus ranges used at different acquiring spots per foil hole to obtain a good resolution-contrast balance (-1.0 to -2.6, -1.0 to -2.8, -1.0 to 3.0 and -1.0 to -3.2). We've now indicated the full correct defocus range of each dataset in Table 1.

3. Regarding the EM reconstruction for KBTBD4:HDAC2 (2:2) in supplementary Figure 1, the map exhibits orientation issues and poor quality. The authors should consider to include 3DFSC or similar metric to describe the anisotropy in data.

We have added the 3DFSC analyses to supplementary Figures 1 and 2.

4. Cryo-EM maps in the figures: please specify what threshold the maps are displayed in the figure legends.

The contour levels for the cryo-EM maps have now been described in the figure legends for Figure 4 and Supplementary Figure 3.

5. Validation of cryo-EM structural models (model-to-map FSC) should be provided.

We have added the model-to-map FSC graphs in the supplementary Figures 1 and 2

6. In terms of deposition, it is recommended that the model for KBTBD4:HDAC2:CoREST1 should be also deposited.

We have built the 2:1:1 structural model as suggested and deposited the coordinates in the PDB (9I2C). Note, due to low local resolutions we have modelled only the backbone coordinates of the CoREST1 subunit. Table 1 has been updated with the model details and the PDB file and validation report have been uploaded for review.

Assays:

7. The cryo-EM structure showed that each HDAC2 molecule interacts with both Kelch domains within the KBTBD4 dimer, while the CoREST1 density is not well resolved, indicating a predominant role for HDAC2 in the CoREST complex binding to KBTBD4. Therefore, it makes sense that CoREST N terminal domain was mapped in Fig 2 as it contains the HDAC2 binding site. From this perspective, it seems that the interaction between KBTBD4 and HDAC2 is independent of CoREST1. What are the cross-linking results when the authors mix KBTBD4R313PRR with HDAC2 alone or CoREST1 alone? Is the affinity between KBTBD4R313PRR and HDAC2 equal to HDAC2-CoREST1? Does CoREST1 have any additional role in complex assembly? In Fig1, please also include IP for HDAC.

We thank the reviewer for this comment regarding the primary role of HDAC2 in the neo-substrate recruitment. We were able to purify full-length HDAC2 alone to perform these experiments. However, CoREST1 could not be purified without HDAC2 due to its poor apparent solubility. As requested, we ran the crosslinking experiment with KBTBD4 and HDAC2 alone and observed a higher molecular weight KBTBD4-HDAC2 complex in the absence of CoREST1 (new Supplementary Figure 6a). Furthermore, HDAC2 alone bound to KBTBD4^{R313PRR} with similar affinity to CoREST1-HDAC2, confirming that HDAC2 drives the complex formation (new Supplementary Figure 6b).

Note, for reviewer #3 we also added new Supplementary Figure 13 showing that the binding affinity using CoREST1-HDAC2 can be increased 2-3 fold in the presence of InsP₆.

As requested, we have also included the IP for HDAC2 in new Figure 1.

8. In Fig3, HDAC2 is shown to be ubiquitinated in KBTBD4R313PRR or KBTBD4WT-UM171 dependent manner, however, the degradation of the other two subunits, CoREST and LSD1 is much greater than that of HDAC1/2, and therefore used in SAR related degradation assay. Could the author explain this observation?

We thank the reviewer for this insightful observation. In cells, HDAC1/2 complexes can incorporate up to 15 different subunits and form assemblies with MW > 1 MDa, whereas our in vitro ubiquitination assay used purified HDAC2 and the truncated CoREST1^{NT} fragment. The accessibility of HDAC2 for ubiquitination may be quite different in these two cases.

We have extended our Discussion section on pages 16/17 to read

“In cells, HDAC1/2 can engage different ELM2-SANT domain proteins to assemble into distinct repressor complexes. In addition to CoREST, UM171 induces some degradation of MIER2, but seemingly not of the NuRD complex containing MTA1 (24). Notably, CoREST and MIER2 are degraded to a greater extent than the direct interactor HDAC1/2, suggesting that in the context of these large multi-subunit complexes HDAC1/2 is less accessible for ubiquitination. ...”

9. What are the affinity values obtained from the individual MST experiment depicted in Fig2c and Fig3b-c?

We have added these values in the related results text on pages 6 and 7, as well as in the figure legends. In Fig. 2c, KBTBD4^{R313PRR} bound to CoREST1^{NT2}-HDAC2 with $K_D = 0.75 \mu\text{M}$. In Fig.3b, KBTBD4^{WT} with 25 μM UM171 bound to CoREST1^{NT2}-HDAC2 more weakly ($K_D \sim 15 \mu\text{M}$). Similarly, in Fig. 3c KBTBD4^{WT} with 5 μM CoREST1^{NT2}-HDAC2 bound to UM171 with $K_D \sim 14 \mu\text{M}$.

10. Have any mutagenesis, in vitro pull down or in vivo IP experiment been done to validate the structural details?

The cryo-EM work is supported in four ways and this is now reflected in new Supplementary Fig. 8 and new discussion text. First, the structures highlighted the critical importance of the cancer mutations in the KBTBD4 BC_II loop (site 1) which has been validated here in our IP experiments and more extensively in our previous publication. Second, the importance of site 2 is confirmed by the monomeric mutant of KBTBD4 used in the cellular IP in Fig. 1. Third, prompted by reviewer #3, we added MST assay data (new Supplementary Fig. 8) showing that HDAC2 binding to KBTBD4^{R313PRR} can be blocked by the HDAC2 inhibitor BRD6929, confirming that the critical importance of the BC_II loop interaction with the HDAC2 catalytic pocket. Fourth, addressing reviewer #3, we have also added a new paragraph in our discussion section to cite the work of Zheng and Liao, who obtained similar structures independently using different methods and constructs and performed systematic base-editing mutagenesis of the proteins.

Minors:

1) In the Method section, "Recombinant protein expression was then induced by addition of 0.4 M isopropyl β -D-1-thiogalactopyranoside", 0.4M or 0.4 mM?

We have corrected this typo to 0.4 mM.

2) The label for Raw blots for Figures 2f should be for Figure 3e.

Thank you for spotting this. We have corrected this mistake.

3) For the raw blots, the labels for protein markers should be included and uncropped version should be provided.

We have updated the uncropped blots to now include the protein markers.

Reviewer #2 (Remarks to the Author)

This is an interesting study that describes and compares the interactions of cancer mutant KBTBD4 with HDAC2 and the UM171-induced complex with HDAC2.

Overall, the manuscript has considerable strengths. It is a well-executed study with some genuinely important findings that make it suitable for publication in Nature Communications. Especially important is the mimicry between the cancer mutations and the effect of the small molecule UM171. I do however have two (related) major concerns that I believe can be addressed without further experiments, but by modifications to both the results and discussion sections.

1. Questionable relevance of the 2:2 structure given that CoREST is not bound to the HDAC2.

Given that the interaction with KBTBD4 is reported to be specific for the CoREST:HDAC complex and yet CoREST is not seen in the complex with mutant KBTBD4, it seems unlikely that the observed 2:2 complex is directly biologically relevant. It is possible that binding to KBTBD4 actively dissociates HDAC2 from the CoREST complex, but that does not fit with the CoREST & LSD1 pull-downs observed here and reported by others.

This does not necessarily mean that the structure is not biologically informative, but this issue needs to be addressed in some detail.

In addressing reviewer #1, we have built the 2:1:1 KBTBD4-CoREST1-HDAC2 structural model and deposited the coordinates in the PDB (9I2C). The addition of this structure has led us to expand our results and discussion text.

For example, results page 11 “The additional binding of CoREST1, however, appears to enforce an asymmetric 2:1:1 assembly due to steric constraints, as noted above (Fig. 4D and Supplementary Fig. 5).”

In the discussion, we now also refer to the preprints of Zheng/Liau that report complementary 2:1:1 structures of KBTBD4-HDAC1-CoREST complexes with the further addition of the HDAC cofactor InsP6, which appears to stabilize both protein-protein interfaces.

2. Specificity for the CoREST complex

The interaction and subsequent degradation of the CoREST:HDAC complex is reported as being rather specific. However the structures do not explain this specificity and this seems to be a major short-coming.

Based on analogy with the HDAC1:MTA1 and HDAC1 MiDAC complexes, and also alphafold predictions of the CoREST:HDAC2 complex, it should be easily possible to predict the disposition of the CoREST proteins with respect to HDAC2. Are these models compatible with the interaction observed for HDAC2 with KBTBD4? Is it possible to explain the specificity for the CoREST:HDAC complex? Or is the HDAC somehow extracted from the complex due to specific properties of the CoREST:HDAC:LSD1 complex?

We have added an additional discussion paragraph to address the specificity question:

“In cells, HDAC1/2 can engage different ELM2-SANT domain proteins to assemble into distinct repressor complexes. In addition to CoREST, UM171 induces some degradation of MIER2, but seemingly not of the NuRD complex containing MTA1 (24). Notably, CoREST and MIER2 are degraded to a greater extent than the direct interactor HDAC1/2, suggesting that in the context of these large multi-subunit complexes HDAC1/2 is less accessible for ubiquitination. Our structures provide opportunity to model the different repressor complexes to gain potential insights into the neomorphic substrate specificity. Modelling of MTA1 (51) predicts a steric clash between its BAH domain and KBTBD4 that may preclude its binding (Supplementary Fig. 5)...”

Reviewer #3 (Remarks to the Author):

The authors present a compelling elucidation of UM171's mechanism of action, including structural insights into the UM171-induced interaction between KBTBD4 and HDAC2. Additionally, they demonstrate a striking structural mimicry between UM171 and a cancer-associated insertion in KBTBD4, as demonstrated by the parallel structural studies of the mutant complex. Additionally, some SAR data of the (now phase II) UM171 compound is presented. Overall, this study is well-executed and clearly described, and in my view it will be of high interest to the journal's broad readership. I am happy to recommend its publication, subject to consideration of the following - mostly minor - suggestions:

1. The authors do not reference a recent bioRxiv preprint from the Liao/Zheng labs that explores the same system and reaches similar (although subtly differing) conclusions. While I do not believe that this preprint in any way precludes publication, it would be prudent to acknowledge this parallel work within the manuscript, especially that largely complementary methodologies were used in the two studies.

We have now cited the two preprints from the Liao/Zheng's labs and described the complementarity of our work as follows:

"In support of our HDAC2-CoREST model, parallel but complementary approaches using HDAC1-CoREST have yielded mutant and UM171-dependent KBTBD4 complexes with a conserved 2:1:1 structural arrangement (preprints 52, 53). An interesting feature of these structures was the inclusion of inositol hexakisphosphate (InsP6), a known HDAC1/2 cofactor. InsP6 binding stabilized the HDAC1-CoREST interface as expected, but also formed a second molecular glue interaction with KBTBD4 to enhance the ternary complex affinity, as we have also observed in our MST assay (Supplementary Fig. 13)."

2. Background interaction between KBTBD4 and HDAC2: The authors mention pre-existing complementarity and present data suggesting potential baseline binding (Fig. 3b, perhaps also 3a) and ubiquitination (Fig. 3e) in the absence of UM171, despite describing some of these as "background noise." Could the authors clarify whether this binding or ubiquitination is significant? E.g. can background interactions or ubiquitination be inhibited with SAHA? If so, is there observable turnover of the complex involving WT KBTBD4 in the baseline state?

Fig. 3b shows that the neo-substrate binding affinity of KBTBD4^{WT} in the absence of UM171 is well above 50 μ M, which is normally perceived as non-significant in the physiological context. We have added to the figure legend the text "...while negligible binding was observed to KBTBD4^{WT} alone". Note, our published cellular IP and MS proteomic data (Pubmed 35379950), revealed no interaction between WT KBTBD4 and HDAC1/2 complexes and no evidence of degradation. A previous study (Chagraoui et al 2021, PMID: 33417871) further showed that KBTBD4 shRNA knock-down in cells did not change the protein level of the LSD1-CoREST1-HDAC2 complex. Thus, the observed surface complementarity between KBTBD4 and HDAC2 at site 2 in our cryo-EM structures appears non-consequential in absence of UM171.

We have added new Supplementary Fig. 8 which shows that the addition of HDAC1/2 inhibitor BRD6929 completely abolishes even the most potent interaction of KBTBD4^{R313PRR} and HDAC2.

3. Role of InsP6: previous studies have suggested a role for InsP6 in HDAC1/2 complexes. Furthermore, the other study proposed that the metabolite, in addition to UM171, is required for ternary complex formation - yet in this study, the complex is observed solely with UM171. Could the authors assess, perhaps via MST or another appropriate method, if InsP6 enhances binding synergistically?

We have assessed the role of InsP6 in both the mutant and UM171 induced complexes using MST assay. We observed that InsP₆ enhanced binding by 2-3 fold and have described this finding in supplementary Fig. 13 and the new discussion paragraph as follows:

“In support of our HDAC2-CoREST model, parallel but complementary approaches using HDAC1-CoREST have yielded mutant and UM171-dependent KBTBD4 complexes with a conserved 2:1:1 structural arrangement (preprints 52, 53). An interesting feature of these structures was the inclusion of inositol hexakisphosphate (InsP6), a known HDAC1/2 cofactor. InsP6 binding stabilized the HDAC1-CoREST interface as expected, but also formed a second molecular glue interaction with KBTBD4 to enhance the ternary complex affinity, as we have also observed in our MST assay (Supplementary Fig. 13).”

4. C2 symmetry with HDAC2 vs asymmetric complex with HDAC1 in the Liao/Zheng lab preprint: Could the authors elaborate on their rationale for processing the structure with C2 symmetry and perhaps comment on the discrepancy between the two studies in how the HDAC1/2 is engaged?

Any symmetry restraint was only applied during the map refinement stage and after the initial map reconstitution. Three distinct species emerged during the data processing for the UM171 cryo-EM dataset, revealing the 2:1 KBTBD4-HDAC2, 2:1:1 KBTBD4-HDAC2-CoREST1 and 2:2 KBTBD4-HDAC2 stoichiometries. These three electron maps were further refined using C1 and C2 symmetries, respectively. The mutant dataset only reconstituted a 2:2 KBTBD4:HDAC2 electron map which was then refined using the C2 symmetry. The different complex stoichiometries in the two studies is now discussed in our revised discussion paragraph including (a) how the inclusion of InsP₆ by Liao/Zheng stabilizes CoREST1 on the associated KBTBD4-HDAC1 complex; and (2) how the stable binding of CoREST1 enforces a 2:1:1 stoichiometry due to a likely steric clash in a putative 2:2:2 model.

We've not seen data to suggest in cells whether HDAC1/2 proteins are always saturated with co-repressor partners, or whether free HDAC1/2 subunits are also present (and whether this varies across different cell types and tissues).

5. For ease of interpretation, it would be beneficial to include the chemical structure of UM171 in Fig 6 (perhaps marking the R groups for the SAR study and this is the figure readers will refer to when reading Table 2)

We have included the chemical structure of UM171 in Fig. 6b. We have kept the schematic scaffold showing the R groups in Table 2, and hope these are most useful for the reader, especially for the R4 substitution position since this is an H atom in the UM171 structure.

6. SAR study: I suggest retesting 2-3 compounds using MST to distinguish effects related to physicochemical properties or cell permeability from effects on ternary complex formation. For the most promising compound, it would be helpful to check for binary interactions with each partner. Also, the observed difference between compounds 1 and 2 is notable—perhaps a retest in an in vitro setup is warranted?

As suggested. we have tested compounds (1) (UM171), (2) (cell inactive) and (13) (increased cellular activity) in the MST assay for ternary complex K_D measurements. These experiments are now described on page 15 of the results as below:

“Using our MST assay, we were able to observe a 3.5-fold improvement in the ternary complex affinity of the most active analog (13)(S234984), whereas there was a 3-fold

reduction for the inactive compound (2), confirming the expected correlation between in vitro binding affinity and cellular activity (Supplementary Fig. 12).

7. Figures:

- a. Fig1a the labels are too close together, which might obscure interpretation

We have edited Fig 1 to clarify the labels.

- b. Fig 5 panels d-f and h-j: these panels could be framed or re-arranged for clarity

We have framed d-f in blue boxes and h-j in grey boxes to help indicate they represent two different binding sites.

- c. Fig. 5-7 or Supplementary Figures: Consider including a comparison of HDAC2 binding an acetyllysine of H3/4 to illustrate the mimicry point

We have included a comparison of acetyllysine vs KBTBD4 Arg312 binding to HDAC2 in the new Supplementary Fig. 8.

- d. Some figures could be condensed, though this is a matter of preference

We have condensed Figures 1 and 2 slightly to minimise white space as suggested.

8. Text:

- a. The opening sentence of the abstract could benefit from a more positive framing

As suggested, we have dropped the negative word “hinder” and edited this sentence to “Neomorphic mutations and drugs can elicit unanticipated effects that require mechanistic understanding to inform clinical practice.”

- b. In the final discussion paragraph, the first two sentences appear to conflate “target” and “ligase” – consider rephrasing to improve readability

We agree. We have edited these sentences to become “To date, only a small number of E3s have been shown to engage a molecular glue. KBTBD4 expands this E3 repertoire and may offer a distinct interaction surface for target substrates less compatible with previously reported E3s.”

NCOMMS-24-58370A. Structural mimicry of UM171 and neomorphic cancer mutants co-opts E3 ligase KBTBD4 for HDAC1/2 recruitment

Response to Reviewers

We thank the reviewers for their approval of our manuscript. We include a point by point response below.

Reviewer #1 (Remarks to the Author):

The authors have addressed the concerns I raised in the previous review. I have one suggestion: all the replicated data should be included in the supplementary file to ensure comprehensive representation.

We thank the reviewer for their time and approval. We have included all the replicated MST assay data in the supplementary source file.

Reviewer #2 (Remarks to the Author):

I am happy that the authors have addressed my concerns and those of the other reviewers. I recommend publication without further revision.

We thank the reviewer for their time and approval.

Reviewer #3 (Remarks to the Author):

The authors have addressed the points raised in my initial review through both additional experiments and revisions to the text and figures. The study remains a well-executed and clearly presented contribution, and I appreciate the effort put into refining it further.

I am happy to recommend its publication in Nature Communications.

We thank the reviewer for their time and approval.